# Short lived organic nitrates in a sub-urban temperate forest: An indication of efficient assimilation of reactive nitrogen by the biosphere?

Simone T. Andersen[1], Rolf Sander[1], Patrick Dewald[1], Laura Wüst[1], Tobias Seubert[1], Gunther N. T. E. Türk[1], Jan Schuladen[1], Max R. McGillen[2], Chaoyang Xue[2], Abdelwahid Mellouki[2,3], Alexandre Kukui[4], Vincent Michoud[5], Manuela Cirtog[5], Mathieu Cazaunau[5], Astrid Bauville[5], Hichem Bouzidi[5], Paola Formenti[5], Cyrielle Denjean[6], Jean-Claude Etienne[6], Olivier Garrouste[6], Christopher Cantrell[7], Jos Lelieveld[1], John N. Crowley[1]

[1]Atmospheric Chemistry Department, Max-Planck-Institute for Chemistry, 55128-Mainz, Germany

[2]Institut de Combustion, Aérothermique, Réactivité Environnement (ICARE), CNRS, 1C Avenue de la Recherche Scientifique, CEDEX 2, 45071 Orléans, France

[3]University Mohammed VI Polytechnic (UM6P), Lot 660, Hay Moulay Rachid Ben Guerir, 43150, Morocco

[4]Laboratoire de Physique et Chimie de l'Environnement et de l'Espace (LPC2E), CNRS Orléans, France

[5]Université Paris Cité and Univ Paris Est Creteil, CNRS, LISA, F-75013 Paris, France

[6]CNRM, Universite de Toulouse, Meteo-France, CNRS, Toulouse, France

[7]Univ Paris Est Creteil and Université de Paris Cité, CNRS, LISA, F-94010 Créteil, France

*Correspondence to*: Simone T. Andersen (simone.andersen@mpic.de) and John N. Crowley (john.crowley@mpic.de)

## 1 Abstract

Alkyl nitrates (ANs) and peroxycarboxylic nitric anhydrides (PANs) are important reservoirs of reactive nitrogen that contribute significantly to the rate of formation and growth of secondary organic aerosols and support the transport of reactive nitrogen from polluted areas to remote areas. It is therefore critical to understand their sources and sinks in different environments. In this study we use measurements of OH, $O_3$, $NO_3$ reactivity, VOCs, $\sum$ANs and $\sum$PANs during the ACROSS (Atmospheric ChemistRy Of the Suburban foreSt) campaign to investigate different production and loss processes of ANs and PANs in a temperate forest. At daytime OH-initiated processes were the dominant source of ANs (69-72 %) followed by $NO_3$ (18-20 %) and $O_3$ (8-12 %). At nighttime the contribution from OH decreased to 43-53 %, and $NO_3$ increased to 26-40 % with that of $O_3$ largely unchanged. Of the measured $\sum$PANs, 48-78 % was modelled to be peroxyacetic nitric anhydride (PAN, commonly known as peroxyacetyl nitrate). Physical loss (e.g. deposition) was an important sink for both ANs and PANs and contributed significantly to the very short lifetimes of 1 - 4 h for ANs and 0.08 - 1.5 h for PANs observed during the campaign.

## 2 Introduction

Approximately 1000 Tg of biogenic volatile organic compounds (BVOCs) are emitted into the atmosphere annually, whereof ~50% is isoprene and ~15% are monoterpenes (Guenther et al., 2012). The oxidation of BVOCs is initiated by ozone ($O_3$) and the hydroxyl (OH) and nitrate ($NO_3$) radicals. When BVOCs are oxidized in the presence of (largely anthropogenic) $NO_x$ they lead to the formation of alkyl nitrates ($RONO_2$, ANs) and peroxycarboxylic nitric anhydrides ($RC(O)O_2NO_2$, PANs) as illustrated in Figure 1. Note that the term PANs includes the most common carboxylic nitrate ($CH_3C(O)O_2NO_2$, commonly referred to as PAN). ANs are produced in the daytime in a minor branch of the reaction between organic peroxy radicals ($RO_2$) which do not have an $\alpha$-carbonyl group and nitrogen oxide (NO, Lightfoot et al. (1992)) and through the reaction of unsaturated BVOCs with $NO_3$ at nighttime. Recent studies suggest that the $NO_3$-initiated oxidation of BVOCs during the day could also be an important pathway for the formation of alkyl nitrates (Liebmann et al., 2019; Ayres et al., 2015; Liebmann et al., 2018a; Liebmann et al., 2018b; Dewald et al., 2024). PANs are formed when $\alpha$-carbonyl peroxy radicals ($RC(O)O_2$) react with nitrogen dioxide ($NO_2$). The stability of PANs is highly temperature dependent, resulting in boundary layer lifetimes of the order of hours at temperate mid-latitudes with respect to dissociation to $RC(O)O_2$ and $NO_2$ (IUPAC, 2024).

The yield of alkyl nitrates in the $RO_2$ + NO reaction strongly depends on the composition of BVOCs in the atmosphere, the oxidant that initiates the degradation of the BVOC (OH, $O_3$ or $NO_3$), and the ratio of NO to other reactants with which the $RO_2$ can react such as $NO_2$, $HO_2$ and other $RO_2$ (IUPAC, 2024; Perring et al., 2013; Wennberg et al., 2018; Hallquist et al., 1999; Fry et al., 2014). The yield of the PANs formed from the corresponding $\alpha$-carbonyl peroxy radical depends only on the fraction of $RC(O)O_2$ that reacts with $NO_2$ as opposed to reacting with $HO_2$ or NO. Production and loss processes of ANs and PANs are described in detail through reaction mechanisms and equations in section 4.1-4.3 and 4.6, respectively.

The formation of ANs and PANs serves to sequester reactive nitrogen ($NO_x$ = NO + $NO_2$) into reservoir species, which can release $NO_x$ following transport to regions remote from $NO_x$ sources; both can also be removed from the lowermost atmosphere through deposition, which thus represents a sink of $NO_x$ in the atmosphere. As $NO_2$ is formed from primary emitted NO

and its photolysis drives the formation of $O_3$, understanding the fate of $NO_x$ is critical for predicting $O_3$ levels in the troposphere. It has additionally been shown that ANs contribute significantly to the rate of formation and growth of secondary organic aerosols (SOA), thereby impacting human health and air quality (Hallquist et al., 2009; Shiraiwa et al., 2017; Kanakidou et al., 2005).

This study investigates the production and loss of PANs and ANs in an anthropogenically impacted temperate forest using field measurements of BVOCs, oxidants (OH, $O_3$, and $NO_3$), products (PANs and ANs), and meteorological data. The lifetimes of PANs and ANs are derived from their production rates and measured total mixing ratios.

### 3    Methodology

### 3.1    Site Description

The measurements used in this study were conducted at the Rambouillet forest site located approximately 50 km southwest of Paris, France, (48.687, 1.704) during the ACROSS (Atmospheric ChemistRy Of the Suburban foreSt) campaign between June 13[th] 2022 and July 25[th] 2022 (Cantrell and Michoud, 2022). The forest consists of approximately 70% oak, 20% pine, and small contributions from beech and chestnut. The top of the forest canopy around the site was approximately 20-25 m. A 40 m measurement tower and multiple containers with a large variety of instruments were located in a roughly quadratic clearing (26.5 m × 26.3 m, ~697 $m^2$) as shown in Figure S1. 48h HYSPLIT back trajectories showed that the airmasses sampled during the campaign passed either over the Atlantic Ocean or Continental regions in Europe before reaching the site (Draxler and Rolph, 2011; Andersen et al., 2024). All the instruments used in this study are described briefly below.

### 3.2    Measurements

### 3.2.1    Reactive Nitrogen and $O_3$

$NO_2$, total peroxycarboxylic nitric anhydrides ($\sum$PANs), total alkyl nitrates ($\sum$ANs), $NO_3$ reactivity ($k^{NO3}$), and $O_3$ were measured by instruments inside the MPIC (Max Planck Institute for Chemistry) container with co-located inlets sampling from a high-volume-flow stainless steel tube (10 $m^3$ $min^{-1}$; 15 cm diameter, 0.2 s residence time) taking air from a height of 5.4 m above ground.

A 5-channel thermal dissociation cavity-ringdown spectrometer (5CH-TD-CRDS, Sobanski et al. (2016)) was used to measure $NO_2$, $\sum$PANs, and $\sum$ANs. $NO_2$ was measured directly at 408 nm with a limit of detection (LOD) of 9.7 parts per trillion by volume (pptv) for 1 min averaging (3σ) and a total uncertainty of 7% + 9.7 pptv + (20 pptv * RH/100). $\sum$PANs and $\sum$ANs were thermally dissociated to $NO_2$ by heating their separate inlets to 448 and 648 K, respectively, followed by detection of $NO_2$ at 408 nm. Numerical simulations were run for both $\sum$PANs and $\sum$ANs to correct for $NO_2$ loss via recombination with $RO_2$, the reactions of peroxy radicals with ambient NO, NO oxidation to $NO_2$ by $O_3$, and pyrolysis of $O_3$ (Thieser et al., 2016; Sobanski et al., 2016). The majority of the correction factors for both $\sum$PANs and $\sum$ANs were between 0.9 and 1.2 as shown in Figure S2, which is consistent with low $NO_x$ levels. The LOD was determined to be 6.3 pptv and 8.6 pptv for $\sum$PANs and $\sum$ANs for 10 min averaging (3σ of the noise on the zero measurements), respectively, and the total uncertainty was 21% +

6.3 pptv and 28% + 8.6 pptv. A timeseries of $NO_2$, $\sum PANs$, and $\sum ANs$ is shown in Figure 2. Two additional cavities, operated at 662 nm, measured $NO_3$ and (via thermal dissociation to $NO_3$, 373 K) $N_2O_5$ (Sobanski et al., 2016). $NO_3$ was throughout the campaign below the LOD of 0.25 pptv for 1 min averaging and $N_2O_5$ was only detected above the LOD of 0.9 pptv on 5 nights and only one of those nights had consistently high $N_2O_5$ for more than an hour. $NO_3$ and $N_2O_5$ data are therefore not presented or used in this study.

A second CRDS-instrument was used primarily to measure the $NO_3$ reactivity towards VOCs in the forest, but it also has a cavity operated at 405 nm for the measurement of $NO_2$ (Liebmann et al., 2018b). The ambient $NO_3$ reactivity was quantified by a CRDS-measurement (at 662 nm) of in-situ-generated $NO_3$ after its residence in a flow-tube reactor when mixed with either synthetic or ambient air. A numerical simulation procedure was used to correct the measurements for competing reactions taking place inside the flow-tube in order to extract the VOC contribution to the measured $NO_3$ consumption. A detailed analysis of the $NO_3$ reactivity measurements is presented in Dewald et al. (2024). During the ACROSS campaign, the reactivity of $NO_3$ towards organics was dominated by those of biogenic origin, so henceforth we refer to this as $k^{BVOC}$.

$O_3$ was measured with a commercial instrument (2B Technologies model 205) using UV absorption at ~254 nm. The LOD is 2 ppbv for 10 s averaging time. A timeseries of $O_3$ can be observed in Figure 2.

NO was measured using a commercial chemiluminescence instrument (Ecophysics CLD 780 TR, henceforth CLD) with an LOD of 10 pptv for 1 min averaging time. The sampling height for NO measurements was about 3.2 m above the ground surface and the inlet was approximately 17 m from the MPIC container at the ICARE-LPC2E container in Figure S1. The NO measurements were corrected for a change in the CLD sensitivity during the campaign caused by an interruption in the instrument's oxygen supply as described in Andersen et al. (2024).

### 3.2.2 OH and $XO_2$ ($HO_2$+$RO_2$)

The OH radical was measured by its conversion (via reaction with isotopically labelled $SO_2$) to $H_2SO_4$ which was subsequently detected using nitrate chemical ionization mass spectrometry (Eisele and Tanner, 1991). The lower limit of detection for OH radicals at signal-to-noise-ratio (S/N) = 3 and a 15-minute integration time was $5\times10^4$ molecule cm$^{-3}$. The sum of peroxy radicals, $XO_2 = HO_2 + RO_2$, was measured by their conversion to OH in the presence of NO. The OH calibration coefficient was determined using $N_2O$ actinometry and OH generation in a turbulent flow reactor by photolysis of $N_2O$ or $H_2O$ at 184.9 nm (Kukui et al., 2008). The calibration of $HO_2$, $CH_3O_2$ and other $RO_2$ was performed by adding into the calibration reactor CO, $CH_4$ (or other $RO_2$ precursors) converting OH to $RO_2$. The overall estimated calibration accuracy ($2\sigma$) for OH is about 25% and about 30% for calibrated $XO_2$, although the uncertainty of the $XO_2$ measurements is typically higher due to variable detection efficiency (i.e. yields of OH) of different $XO_2$. Peroxy radicals derived from terpene oxidation will be detected with a lower efficiency than e.g. short chained $RO_2$ so that the $XO_2$ measurements during ACROSS should be regarded as lower limits. The lower limit of detection for $XO_2$ radicals at S/N=3 and a 4-minute integration time is $2\times10^6$ molecule cm$^{-3}$. The background signal is determined by adding OH and $XO_2$ scavenger, $NO_2$ in this case,

before their conversion to $H_2SO_4$. As discussed in Kukui et al. (2021), formation of $H_2SO_4$ in the reaction of stabilized Criegee intermediates (SCI) with $SO_2$ in the conversion reactor may lead to some positive interference in radical measurements. However, this interference estimated to correspond to several $10^4$ $cm^{-3}$ of radicals concentrations is negligible compared to measured OH concentrations. A detailed description of the instrument and calibration system have been presented elsewhere (Kukui et al., 2008; Kukui et al., 2021). During the ACROSS field campaign the instrument was installed in the ICARE-LPC2E shipping container with the chemical conversion reactor fixed to the roof of the container via an interface cap covered with a polytetrafluoroethylene (PTFE) sheet. The sampling aperture of the chemical conversion reactor (3 mm diameter) was positioned 50 cm above the roof and about 3 m above the ground surface. A detailed analysis of the OH measurements will be presented in a forthcoming publication.

### 3.2.3   Photolysis frequencies and meteorology

Spectral radiometers (Metcon Gmbh) were installed near the high-volume-flow stainless-steel tube on top of the MPIC container and on top of the tower to measure actinic fluxes, which were converted to photolysis frequencies using recommended absorption spectra and quantum yields (IUPAC, 2024; Burkholder et al., 2020) as described in Meusel et al. (2016). Note that upwelling radiation is not accounted for resulting in a potential underestimation of the photolysis frequencies of 5-10 %. A comparison of the two measurement heights is shown in Figure S3. It is clear that the measurements below the canopy are significantly impacted by shading from trees in the morning and afternoon as well as by the tower around midday. Differences between the two measurements when both instruments are in direct sunlight is caused by integration of solar flux over one complete hemisphere (basically above the height of the integrating dome), which is influenced by trees at angles close to horizontal.

Ambient temperature was measured at four different heights on the tower; 5 m, 13 m, 21 m, and 41 m using temperature sensors from Atexis (PT1000) and Thermoest (PT100). Relative humidity was measured at 5 m using a Vaisala humidity sensor (HMP45A).

### 3.2.4   Biogenic Volatile Organic Compounds (BVOCs)

The LISA (Laboratoire Interuniversitaire des Systèmes Atmosphériques) Proton Transfer Reaction Time of Flight Mass Spectrometer (PTR-ToF-MS, hereafter called PTRMS), manufactured by Kore Technology Ltd., was used for monitoring concentrations of VOCs. Air samples were drawn at a flow rate of approximately 300 mL $min^{-1}$ through a 3 m long Silcosteel® coated stainless steel tube (2.1 mm inner diameter) from an inlet height of 4.6 m in the LISA observation container in Figure S1. Calibration was performed approximately every 3 days using VOC standards (5 – 20 ppb) from a certified National Physical Laboratory (NPL) calibration mixture with nominally 1 ppmv ±5% of several trace gases including acetaldehyde, methanol, ethanol, isoprene, acetone, dimethyl sulphide, acetonitrile, and 3-carene. Humidity corrections were applied for each trace-gas. Mixing ratios of non-calibrated trace gases were retrieved from reactor conditions, rate constants, fragmentations, and ion transmissions determined using the same NPL standard cylinder. The time series of acetaldehyde, isoprene and total monoterpenes ($\sum$MT) is shown in Figure 2.

As there were no measurements of speciated monoterpenes due to instrumental issues with the gas chromatography instrument deployed during the campaign, different potential monoterpene mixtures were determined using the measured reactivity of $NO_3$ towards BVOCs ($k^{BVOC}$) after subtracting the reactivity due to isoprene ($k_{NO3+isoprene}[Isoprene]$) as described in Eq. (1). Here, $k_{effective}$ is an effective rate coefficient for the reaction of $NO_3$ with an assumed monoterpene mixture. $k_{effective}$ was determined by adding all the fractional contributions ($a_i$) from different monoterpenes as described in equation (2), where $k_{NO3+i}$ is the rate coefficient with $NO_3$ for monoterpene $i$, which are listed in Table 1. The estimated $\sum MT$ from equation (1) using different $k_{effective}$ can then be compared to the total mixing ratios measured by the PTRMS.

$$[\sum \text{Monoterpenes}] = \frac{k^{BVOC} - k_{NO3+isoprene}[Isoprene]}{k_{effective}} \qquad (1)$$

$$k_{effective} = \sum_i (a_i \times k_{NO3+i}) \qquad (2)$$

Only limonene, α-pinene, and β-pinene were used to determine the potential mixtures since the box-model used (see below) contained schemes for their degradation only. Four examples of potential mixtures constrained by the measured $NO_3$-reactivity and consistent (for parts of the day) with the measured sum of monoterpenes ($\sum MT$) from the PTRMS were determined by varying the fractional contribution ($a_i$) of limonene between 0 and 30%. Mixture 1: 30% limonene, 10% β-pinene, and 60% α-pinene. Mixture 2: 20% limonene, 15% β-pinene, and 65% α-pinene. Mixture 3: 10% limonene, 5% β-pinene, and 85% α-pinene. Mixture 4: 57% β-pinene and 43% α-pinene. All four scenarios are plotted together with the measured $\sum MT$ over a 48-hour period in Figure 3B. Here the three scenarios, which include limonene, can be observed to agree well with the measurements except when a temperature inversion occurred (Figure 3A). The same kind of discrepancy is observed every night with a temperature inversion, which is shown in Figure 3C, where mixture 2 is plotted against the measured $\sum MT$ for the entire campaign and coloured by the difference in temperature ($\Delta T$) between the top of the tower (41 m) and measurement height (5m). A significant temperature inversion can result in the formation of a shallow nocturnal surface layer with weak vertical mixing. In contrast, the fourth mixture, which does not include limonene, does agree well with the measured $\sum MT$ when a temperature inversion is observed, but it would require a much larger (than measured by the PTRMS) total mixing ratio of monoterpenes when no temperature inversion is observed.

As BVOCs continue to be emitted at night, weak vertical mixing leads to a strong gradient in monoterpene mixing ratios with higher values at low heights above ground level. As the gradient (both horizontally and vertically) will depend on the lifetime of each monoterpene, the mixture of monoterpenes measured by the $NO_3$-reactivity instrument (at 5.4 m) might not be the same as that emitted by the vegetation at different heights. In Figure S4, the average derived diel profiles of the lifetime of α-pinene, β-pinene, and limonene when taking reactions with OH, $O_3$, and $NO_3$ into account using the rate coefficients in Table 1 are plotted. During the daytime all three monoterpenes have short lifetimes of around 0.5-1.5 hours; however, at nighttime, limonene clearly has the shortest lifetime, whereas β-pinene has a slightly longer lifetime than α-pinene. On nights with $\Delta T>1°C$ (26% of the total measurements between June 17th 2022 and July 22nd 2022), limonene is therefore assumed not to be sampled by the $NO_3$-reactivity instrument due to the slow vertical and horizontal mixing and monoterpene mixture 1-3 are changed to 57% β-pinene and 43% α-pinene (Mixture 4), which can be seen in Figure

3D for Mixture 2. This correction aligns the calculated and measured $\sum$MT throughout the entire campaign (Figure 3E). It should also be noted that the $NO_3$-reactivity and PTRMS measurements were not co-located and there could therefore also be a gradient between the measurements, which we do not take into account as the magnitude of this gradient is unknown.

### 3.3   Box Model

To simulate PANs numerically, we have used the atmospheric chemistry box model CAABA/MECCA (Chemistry As A Boxmodel Application/Module Efficiently Calculating the Chemistry of the Atmosphere) by Sander et al. (2019). The code is based on model version 4.7.0, and it has been adapted to simulate the ACROSS campaign. To allow for a detailed calculation of monoterpenes and PANs, reactions were exported from the Master Chemical Mechanism (MCM, https://mcm.york.ac.uk), including the MCM species APINENE, BPINENE, $C_3H_8$, $C_5H_8$, $CH_3CHO$, $CH_4$, LIMONENE, and $NC_4H_{10}$ in the marklist. This resulted in a gas-phase chemical mechanism with 1536 species and 4550 reactions. The setup of the individual model runs will be described in Sect. 4.7.

### 4   Results and Discussion

### 4.1   AN production from $NO_3$ reactions with BVOCs

$NO_3$ radicals are produced from the reaction between $NO_2$ and $O_3$ (R1) and are usually lost rapidly during daytime to photolysis and reactions with unsaturated BVOCs and NO (R2-R4). Reactions between $NO_3$ radicals and unsaturated BVOCs lead to the formation of alkyl nitrates (ANs, R3a) as well as other products (R3b). At nighttime, ground-level $NO_3$ mixing ratios can vary greatly from < 1 pptv to > 100 pptv depending on atmospheric composition (Ng et al., 2017; Brown and Stutz, 2012). During ACROSS, the $NO_3$ reactivity within the canopy was generally high at nighttime due to the emission of biogenic volatile organic compounds (BVOCs, by vegetation) and NO (from soil) into a shallow, poorly mixed nocturnal surface layer (Dewald et al., 2024; Andersen et al., 2024). This led to $NO_3$ mixing ratios, at the ground, of < 0.5 pptv, which were generally lower than the limit of detection of instrumentation at the site.

$$NO_2 + O_3 \rightarrow NO_3 + O_2 \tag{R1}$$

$$NO_3 + h\nu \rightarrow NO_2 + O \tag{R2a}$$

$$NO_3 + h\nu \rightarrow NO + O_2 \tag{R2b}$$

$$NO_3 + BVOC\ (O_2, NO, RO_2, HO_2) \rightarrow ANs \tag{R3a}$$

$$NO_3 + BVOC \rightarrow other\ products \tag{R3b}$$

$$NO_3 + NO \rightarrow 2\ NO_2 \tag{R4}$$

The total production rate of ANs from the $NO_3$-initiated oxidation of unsaturated BVOCs can be calculated using equation (3), where $[NO_3]_{SS}$ is the $NO_3$ concentration at steady state and $\alpha_i^{NO_3}$, $k_i^{NO_3}$ and $[C_i]$ are the ANs yield, the rate coefficient and BVOC concentration for compound $i$, respectively. $[NO_3]_{SS}$ is determined from the production and loss terms described by reactions (R1-R4) as described in equation (4), where [NO], $[NO_2]$, and $[O_3]$ are the

concentrations of NO, NO$_2$, and O$_3$, respectively, $k_1$ and $k_4$ are the rate coefficients of reaction (R1) and (R4), respectively, $k^{\text{BVOC}}$ is the first-order loss frequency for NO$_3$ towards BVOCs, and $J_{\text{NO3}}$ is the photolysis frequency of NO$_3$ radicals (Liebmann et al., 2019). This calculation ignores physical losses of NO$_3$ (e.g. deposition) which will not compete with its reactive losses in this environment.

$$\sum P_{\text{ANs}}^{\text{NO3}} = [NO_3]_{\text{SS}} \sum_i \alpha_i^{\text{NO}_3} k_i^{\text{NO}_3} [C_i] \tag{3}$$

$$[NO_3]_{\text{SS}} = \frac{k_1[NO_2][O_3]}{k^{\text{BVOC}} + J_{\text{NO3}} + k_4[NO]} \tag{4}$$

## 4.2 AN production from OH reactions with BVOCs

At daytime, primary OH radicals are produced e.g. from the photolysis of O$_3$ followed by the reaction between O($^1$D) and water vapour (R5-R6) with secondary production through reaction of HO$_2$ (formed in peroxy-radical (RO$_2$) reactions) with NO. In the absence of photochemistry, OH concentrations are generally lower at nighttime than at daytime, with average hourly concentrations of around $3.5\text{-}5 \times 10^5$ molecules cm$^{-3}$ observed across the campaign. The most important nighttime source of OH radicals is generally believed to be the reaction between unsaturated VOCs and ozone e.g. (R11-R12). As will be shown in a forthcoming publication, the nighttime OH levels are broadly consistent with measured OH-reactivity, HO$_2$-recycling via reaction of HO$_2$ with NO and production via the ozonolysis of terpenoids. When OH reacts with BVOCs in the presence of O$_2$, peroxy radicals are produced (R7), which can then react with NO to give alkyl nitrates (R8a), as well as alkoxy radicals and NO$_2$ (R8b). Additional competing processes that lower the yield of ANs from RO$_2$ are reactions with itself and other RO$_2$ (R9) and reaction with HO$_2$ (R10).

$$O_3 + h\nu \rightarrow O(^1D) + O_2 \tag{R5}$$

$$O(^1D) + H_2O \rightarrow 2\ OH \tag{R6}$$

$$OH + BVOC\ (+O_2) \rightarrow RO_2 \tag{R7}$$

$$RO_2 + NO + M \rightarrow RONO_2 + M \tag{R8a}$$

$$RO_2 + NO \rightarrow RO + NO_2 \tag{R8b}$$

$$RO_2 + RO_2 \rightarrow products \tag{R9}$$

$$RO_2 + HO_2 \rightarrow ROOH + O_2 \tag{R10}$$

The total production rate of ANs from OH-initiated oxidation of BVOCs is described in equation (5), where [OH] is the OH concentration, $\alpha_i^{\text{RO}_2 + \text{NO}}$ is the fraction of the organic peroxy radicals from BVOCs which (via R8a) forms an alkyl nitrate when reacting with NO, $k_i^{\text{OH}}$ and [C$_i$] are the rate coefficient and BVOC concentration for compound $i$, respectively, and $\beta$ is the fraction of peroxy radicals that reacts with NO (rather than RO$_2$ or HO$_2$) as described in equation (6) (Liebmann et al., 2019). $\beta$ was calculated using the measured XO$_2$ (HO$_2$+RO$_2$) and NO together with a generic rate coefficient for reaction (R8), $8 \times 10^{-12}$ cm$^3$ molecule$^{-1}$ s$^{-1}$, and a generic rate coefficient for the combination of reactions (R9) and (R10), $k_{9/10} = 1 \times 10^{-11}$ cm$^3$ molecule$^{-1}$ s$^{-1}$ (IUPAC, 2024; Lightfoot et al., 1992). $k_8$ was set to this value because the vast majority of organic peroxy radicals react with NO with a rate coefficient

of $8 \pm 1 \times 10^{-12}$ cm$^3$ molecule$^{-1}$ s$^{-1}$ (IUPAC, 2024; Lightfoot et al., 1992). Since we do not have separate measurements of RO$_2$ and HO$_2$ an effective rate coefficient of $1 \times 10^{-11}$ cm$^3$ molecule$^{-1}$ s$^{-1}$ was chosen. This was derived by considering the MCM rate coefficients ($2.3 \times 10^{-11}$ cm$^3$ molecule$^{-1}$ s$^{-1}$ at 298 K) for reaction (R10) and $2\text{-}3 \times 10^{-12}$ cm$^3$ molecule$^{-1}$ s$^{-1}$ for reaction between isoprene derived peroxy radicals and other RO$_2$ and calculating the geometric mean. As we do not know the contributions of HO$_2$ and RO$_2$ to XO$_2$ this is clearly a coarse approximation. We therefore undertook a sensitivity test in which the effective rate coefficient was varied by a factor of two in both directions to establish the uncertainty in $\beta$ using median mixing ratios across the entire campaign of NO (41 pptv) and XO$_2$ ($6 \times 10^8$ molecule cm$^{-3}$). When halving the effective rate coefficient $\beta$ is increased by 30% and thereby increasing the production rate and when doubling the effective rate coefficient $\beta$ is decreased by 30% and thereby decreasing the production rate.

$$\sum P_{ANs}^{OH} = [OH]\beta \sum_i \alpha_i^{RO_2+NO} k_i^{OH}[C_i] \tag{5}$$

$$\beta = \frac{k_8[NO]}{k_8[NO]+k_9[RO_2]+k_{10}[HO_2]} = \frac{k_8[NO]}{k_8[NO]+k_{9/10}[XO_2]} \tag{6}$$

### 4.3 AN production from O$_3$ reactions with BVOCs

Ozone (O$_3$) addition to an unsaturated BVOC forms a primary ozonide (POZ, R11), which in the presence of O$_2$ can rapidly decompose via Criegee intermediates to OH and organic peroxy radicals (RO$_2$) (R12a). The POZ can also react through other processes that do not result in organic peroxy radicals (R12b). The RO$_2$ formed in (R12a) further reacts through reactions (R8-R10) as described above.

$$O_3 + R\text{=}R \rightarrow POZ \tag{R11}$$

$$POZ (+O_2) \rightarrow OH + RO_2 \tag{R12a}$$

$$POZ \rightarrow \text{other products} \tag{R12b}$$

Equation (7) describes the total production rate of ANs from the O$_3$-initiated oxidation of unsaturated BVOCs, where [O$_3$] is the O$_3$ concentration, $\alpha_i^{O_3}$ is the yield of RO$_2$ from reactions (R11-R12a), $\alpha_i^{RO_2+NO}$ is the fraction of the organic peroxy radicals (from BVOCs) formed in (R12a) which when reacting with NO form an alkyl nitrate, $k_i^{O_3}$ and $[C_i]$ are the rate coefficient and BVOC concentration for compound $i$, respectively, and $\beta$ is the fraction of peroxy radicals that react with NO as described in equation (6) (Liebmann et al., 2019), calculated as described in section 4.2. Here we assume that the yield of ANs from RO$_2$ + NO is independent of whether RO$_2$ is formed by OH or O$_3$-initiated oxidation.

$$\sum P_{ANs}^{O_3} = [O_3]\beta \sum_i \alpha_i^{O_3} \alpha_i^{RO_2+NO} k_i^{O_3}[C_i] \tag{7}$$

### 4.4 Relative importance of OH, O$_3$ and NO$_3$ oxidation for the production of ANs

To analyse the production and loss processes of alkyl nitrates during the ACROSS campaign, the measurements have been separated into two phases: phase 1 (lower photochemical activity) from June 28$^{th}$ 2022 to July 7$^{th}$ 2022, and phase 2 (higher photochemical activity) from July 8$^{th}$ 2022 to July 20$^{th}$ 2022. Average diel profiles of O$_3$, NO$_2$, NO, temperature, OH radicals, and

isoprene for the two phases are plotted in Figure 4. Phase 1 is characterised by low levels of oxidants and organics and a maximum average daytime temperature around 25 degrees Celsius, whereas phase 2 is characterised by higher levels of oxidants and organics due to generally higher temperatures (maximum average daytime temperature around 30 degrees Celsius). The average monoterpene mixtures for the two phases are displayed in Figure 5 for the mixture

with 20% limonene, 15% β-pinene, and 65% α-pinene when no temperature inversion is observed and 57% β-pinene and 43% α-pinene when temperature inversions higher than 1°C is observed. The monoterpene mixtures derived for Mixture 1 and 3 using 57% β-pinene and 43% α-pinene when temperature inversions higher than 1°C is observed  are plotted in Figure S5. At daytime the average mixture reflects the percentages used for each scenario, whereas at

nighttime, the mixture is dominated by β-pinene and α-pinene during both phases, associated with the temperature inversions observed on many nights.

The diel profile of the $\sum$ANs production rates from $NO_3$-, OH-, and $O_3$-initiated oxidation using the rate coefficients and yields in Table 1 and the monoterpene mixture in Figure 5 is shown in Figure 6A and B for phases 1 and 2, respectively. The total ANs production during phase 1 is

fairly constant at 35-75 pptv h$^{-1}$ throughout the diel profile. In contrast, phase 2 shows large variation with 80-100 pptv h$^{-1}$ in the early morning between 01:00-05:00 UTC (03:00-07:00 LT) and 220-280 pptv h$^{-1}$ around late morning to midday between 07:00-12:00 UTC (09:00-14:00 LT). For both phases, the variability in the $\sum$ANs production rate from $O_3$ and $NO_3$-initiated oxidation of BVOCs is small compared to that of OH-initiated oxidation. The

calculated $O_3$-initiated $\sum$ANs production rate varied from 4-15 and 12-25 pptv h$^{-1}$ for phase 1 and 2, respectively, and the derived $NO_3$-initiated $\sum$ANs production rate varied from 4-22 and 22-62 pptv h$^{-1}$ for phase 1 and 2, respectively. The differences between the two phases are therefore caused by the OH-initiated oxidation, which is relatively stable during phase 1 at 13-61 pptv h$^{-1}$ across the diel profile due to the lower daytime levels of OH and BVOCs (see

Figure 4), but varies between 30-70 pptv h$^{-1}$ at nighttime and 100-230 pptv h$^{-1}$ at daytime during phase 2. The production rates are associated with uncertainties, which have been evaluated using the average midday measurements and derived monoterpene mixing ratios in table S1, the uncertainties in rate coefficients (see table 1 and S2), measurement uncertainties and yield uncertainties (see table S2) for the three monoterpene mixtures containing limonene. The

results are summarized in table S3 and show that the uncertainty varies only a little within the different monoterpene mixtures. At midday the propagated uncertainty of the $NO_3$-, $O_3$-, OH-initiated production rate, and the total production rate is around 35%, 40%, 30%, and 26%, respectively.

Figure 6C and D show the fractional contributions of $NO_3$-, OH-, and $O_3$-initiated oxidation to

the $\sum$ANs production rate for phase 1 and 2, respectively. OH clearly dominates at daytime (06:00-18:00 UTC) with, on average, 69-72 % for both phases, followed by $NO_3$ with 18-20 % and $O_3$ with 8-12 %. At nighttime (18:00-06:00 UTC), the picture is not as clear: During phase 1 OH-initiated oxidation is still the dominant ANs production pathway with, on average, 53 % and the remainder is close to evenly split at 21 and 26 % between $O_3$- and $NO_3$-initiated

oxidation, respectively. At nighttime during phase 2, OH- and $NO_3$-initiated oxidation contributed similarly with 43 and 40 % of the total ANs production rate, respectively, leaving only 17 % for $O_3$-initiated processes. All the fractional contributions are associated with uncertainties around 40-50% when propagating the uncertainties in the individual and total production rates at midday. The differences between the two phases at nighttime can be

explained by the availability of the precursors, where there is approximately double the amount of $O_3$ and $NO_2$ during phase 2, leading to a higher production rate of $NO_3$ radicals and thereby an increased ANs production rate from $NO_3$-initiated oxidation.

Both phases give significantly different fractions at both daytime and nighttime from those observed by Liebmann et al. (2019) in a boreal forest, where, in the absence of measurements,
OH was calculated from the actinic flux, which thus resulted in zero OH at nighttime. However, both studies agree on $NO_3$ oxidation being a significant source of ANs, both at daytime and nighttime. If $[NO_3]_{SS}$ was calculated using photolysis frequencies measured above the clearing instead of inside the clearing, the contribution from $NO_3$-initiated oxidation would be reduced in the morning and evening (see Figure S3 for comparison of $J_{NO3}$ above (tower) and inside
(ground) the clearing). Further as discussed by (Dewald et al., 2024) the relative contributions of e.g. OH and $NO_3$ would be significantly modified to favour $NO_3$ if we consider the greatly reduced photolysis frequencies of $NO_3$ and OH precursors in non-cleared parts of the forest.

### 4.5    ANs loss and lifetime
Neglecting the role of transport, we now combine the diel profile of the total production rate of ANs (described above) with the average measured diel profile of ANs to evaluate the loss processes and lifetime of the ANs using equation (8). $P_{ANs}$ is the production rate of ANs, $[ANs]_0$ is the average ANs mixing ratio at 00:00 UTC, and $k_L(ANs)$ is the loss rate of the ANs, which is defined as the inverse of the lifetime of the ANs ($(\tau_{ANs})^{-1}$).

$$\frac{d[ANs]}{dt} = P_{ANs} - k_L(ANs)[ANs]_0 \tag{8}$$

The ANs mixing ratio at any subsequent time to $[ANs]_0$ can then be calculated as described in Eq. (9) with the variation of $k_L(ANs)$ to match the observed ANs mixing ratio.

$$[ANs]_t = \int_0^t \frac{d[ANs]}{dt} + [ANs]_0 \tag{9}$$

Figure 6E and F show the average diel profiles of the measured ANs in black for phase 1 and
2, respectively. The orange lines show how the ANs mixing ratios would have increased if there was no chemical or transport-induced loss throughout the day and the blue lines show how the diel profiles would look when applying lifetimes of 1-10 hours for the ANs. The best fit to the measured diel profile of ANs results from using an effective lifetime in the clearing of $1.5 \pm 1$ h throughout the diel cycle for both phases despite the very different production rates
described above. Table 2 gives an overview of the average daytime and nighttime loss rate frequencies and the resulting effective lifetimes. No difference was observed between daytime and nighttime during phase 1 and only a small difference within the uncertainties was observed during phase 2.

The total ANs production rate resulting from the three different monoterpene mixtures derived
in section 3.2.4 (plotted in Figure S6A for the two phases) clearly shows that the choice of mixture does not have a significant impact on the total production rate. This means that the determined lifetime for the two phases of $1.5 \pm 1$ h is consistent across all three mixtures, as can be observed in Figure S6C. This lifetime is similar to the $2 \pm 3$ h estimated in a boreal forest (Liebmann et al., 2019) and that determined for isoprene nitrates using flux
measurements over the Ozark Mountains (Wolfe et al., 2015).

In section 4.7, it is shown that the box model described in Section 3.3 predicts the mixing ratio of $XO_2$ to be up to 4 times higher than the measured $XO_2$ depending on the day, which is related in part to the variable detection efficiency of different $RO_2$ by the $XO_2$ instrument. Strictly speaking the $XO_2$ measurements represent a lower limit in regions with high biogenic contribution to $RO_2$. Owing to the large uncertainty associated with the measurements we have also calculated the total ANs production rate and the lifetime of ANs for the three monoterpene mixtures when applying 4 times the measured $XO_2$. The results (Figures S6B and S6D) show that $\beta$ and thus $P_{ANs}$ decrease when increasing $XO_2$, which results in an increase in the calculated lifetime from $1.5 \pm 1$ h to $2.5 \pm 1.5$ h, which remains consistent with the previous observations described above.

### 4.6 PANs major production and loss processes

A dominant fraction of the measured PANs is expected to be in the form of peroxyacetic nitric anhydride ($CH_3C(O)O_2NO_2$, also known as peroxyacetyl nitrate, PAN), which is formed in the reaction between the peroxyacetyl radical ($CH_3C(O)O_2$) and $NO_2$ (R13). In the boundary layer, $CH_3C(O)O_2$ is produced directly from the OH-initiated oxidation of acetaldehyde ($CH_3CHO$, R14) and the photolysis of dicarbonyls such as methylglyoxal ($CH_3C(O)CHO$, R15, Crowley et al. (2018)) and from the oxidation of BVOCs such as isoprene (via methacrolein ($CH_2C(CH_3)CHO$, MACR) and methyl vinyl ketone ($CH_2CHC(O)CH_3$, MVK)) and α-pinene after multiple reaction steps.

$$CH_3C(O)O_2 + NO_2 \quad \rightarrow \quad CH_3C(O)O_2NO_2 \text{ (PAN)} \quad \text{(R13)}$$

$$OH + CH_3CHO + O_2 \quad \rightarrow \quad CH_3C(O)O_2 + H_2O \quad \text{(R14)}$$

$$CH_3C(O)CHO + h\nu + 2\,O_2 \quad \rightarrow \quad CH_3C(O)O_2 + CO + HO_2 \quad \text{(R15)}$$

In forest environments in the summer, when isoprene emissions are high, peroxymethacrylic nitric anhydride ($CH_2C(CH_3)C(O)O_2NO_2$, MPAN), an OH-initiated oxidation product of MACR (R16-R17), will contribute to the total peroxy nitrates. Note that $CH_2C(CH_3)C(O)O_2$ can also isomerize (Crounse et al., 2012), which is not included in the MCM. With an annual average propane ($C_3H_8$) measurement around 507 pptv at La Tardiere, France, in 2018 (Ge et al., 2024), peroxypropionic nitric anhydride ($CH_3CH_2C(O)O_2NO_2$, PPN), an OH-initiated oxidation product of propanal ($CH_3CH_2CHO$, R18-R19), is expected to be present as well.

$$OH + CH_2C(CH_3)CHO \text{ (MACR)} + O_2 \quad \rightarrow \quad CH_2C(CH_3)C(O)O_2 + H_2O \quad \text{(R16)}$$

$$CH_2C(CH_3)C(O)O_2 + NO_2 \quad \rightarrow \quad CH_2C(CH_3)C(O)O_2NO_2 \text{ (MPAN)} \quad \text{(R17)}$$

$$OH + CH_3CH_2CHO + O_2 \quad \rightarrow \quad CH_3CH_2C(O)O_2 + H_2O \quad \text{(R18)}$$

$$CH_3CH_2C(O)O_2 + NO_2 \quad \rightarrow \quad CH_3CH_2C(O)O_2NO_2 \text{ (PPN)} \quad \text{(R19)}$$

For both PAN, MPAN, PPN and any other PANs, the production rate strongly depends on the concentration of NO, hydroperoxyl radicals ($HO_2$) and other peroxy radicals ($RO_2$) that can lead to competing reactions (R20-R21) to e.g. (R13), (R17) and (R19). $XO_2$ represents the sum of $HO_2 + RO_2$.

$$RC(O)O_2 + NO \quad \rightarrow \quad R + CO_2 + NO_2 \quad \text{(R20)}$$

$RC(O)O_2 + XO_2$        $\rightarrow$        $RC(O)O + XO + O_2$             (R21)

PANs are permanently removed through deposition (R22) and can be lost through thermal decomposition (R23) reforming $RC(O)O_2$, which can subsequently react as described above (R20-R21). Larger and/or unsaturated PANs, such as MPAN, can also be lost through oxidation (R24).

$RC(O)O_2NO_2$                $\rightarrow$        Deposition                  (R22)

       $RC(O)O_2NO_2$                $\rightarrow$        $RC(O)O_2 + NO_2$           (R23)

       $RC(O)O_2NO_2 + OH$        $\rightarrow$        Products                 (R24)

The thermal decomposition of PANs is strongly temperature dependent resulting in lifetimes with respect to (R23) from 7.5 hours at 283 K and 40 minutes at 298 K (IUPAC, 2024). The
495 effective lifetime increases from that calculated from the thermal decomposition rate coefficient when regeneration of PANs through e.g. reaction (R13), (R17), and (R19) occurs. Thermal decomposition is thus expected to be the dominant loss process of PANs at high temperatures in the presence of NO and/or $XO_2$, however, at nighttime, when the temperature is lower and the mixing ratios of NO and $XO_2$ also are lower, deposition can play an important
role, depending on boundary layer height, humidity and surfaces.

### 4.7    Measured and modelled PANs

Figure 7A plots the measured mixing ratio for $\Sigma$PANs (=PAN + MPAN + PPN + other PANs) for which maximum daytime mixing ratios are between 100 pptv and 1600 pptv. The large
variability is presumably caused by the observed variability in temperature, concentrations of oxidants, and BVOCs. The temperature fluctuations measured during the campaign result in a thermal lifetime of the PANs that spans two orders of magnitude, from 15 hours at 279 K to 3 minutes at 314 K as shown in Figure 7B.

Due to the many different production pathways of PANs, we used a detailed chemical box-
510 model (see section 3.3 for details) to assess the contributions of various precursors and compare to measured $\Sigma$PANs. As the calculated thermal loss rate varies significantly from day to day, two single days (marked in grey in Figure 7), where measurements of OH, $O_3$, NO, $NO_2$, and BVOCs are available, have been modelled instead of using average diel profiles as in the ANs analysis. One day (July 4th 2022) is in phase 1 from the ANs analysis, where the temperature
reaches around 27°C (300 K) resulting in a thermal lifetime of 25 minutes (without considering recombination) and during which the mixing ratios of precursors (oxidants and BVOCs) were low. The second day (July 13th 2022) is in phase 2 from the ANs analysis, where the temperature has a maximum of around 39°C (312 K) resulting in a thermal lifetime of around 5 minutes (without considering recombination) and higher precursor levels than during the first
520 day.

The box model was constrained by measurements (20-minute running averages) of temperature, humidity, $JNO_2$, $J(O^1D)$, OH, $O_3$, NO, $NO_2$, isoprene, and acetaldehyde ($CH_3CHO$), and the prescribed 20% limonene monoterpene mixture described above. Note that varying the fraction of limonene did not change the modelled $\Sigma$PANs significantly. The model
was additionally initiated using average CO and pressure for the modelled day, 507 pptv

propane ($C_3H_8$), 193 pptv n-butane (n-$C_4H_{10}$), and 107 pptv isobutane (i-$C_4H_{10}$) (Ge et al., 2024). To ensure that all the trace gases not constrained in the model were in steady-state, a spin-up time of nine days was used. The thermal losses of PANs were included in the model as described above and an additional first-order loss term (physical losses from e.g. deposition) was varied in two separate model runs to get satisfactory model-measurement agreement for ∑PANs during daytime (06:00-18:00 UTC) and nighttime (18:00-06:00 UTC) for the individual days. The PANs identified in the MCM are listed in table S4 and the fractional contributions of PAN, MPAN, and PPN to the total are described in the supplementary information.

*Modelled loss rates optimized to match nighttime observations of PANs:* The modelled PANs, separated into PAN and other PANs when optimized for nighttime are plotted for the low and high precursor day in Figure 8A and B together with the measured mixing ratios. Reasonable agreement between the measured and modelled ∑PANs can be observed for both the high and low precursor day at nighttime, but overestimates part of the day. The daytime was therefore modelled separately (see below).

The physical loss frequencies required to align the modelled ∑PANs at nighttime with the observations were $1.8 \times 10^{-4}$ $s^{-1}$ and $1.1 \times 10^{-3}$ $s^{-1}$ for the low and high precursor day, respectively. By comparison the nighttime loss frequency attributable to thermal decomposition when taking recombination into account (at temperatures of 10-20 °C, see Figure 8C and D) is negligible. The resulting average nighttime lifetime (1.5 and 0.24 h, see Table 2) is therefore determined almost solely by physical losses.

*Modelled loss rates optimized to match daytime observations of PANs:* When optimizing for daytime agreement in the model (see Figure S7A and B) reasonable agreement can be observed between 06:00-18:00 for the high precursor day. For the low precursor day, the model overestimates the measurement between 06:00 and 12:00 UTC and underestimates between 12:00 and 18:00 UTC. The physical loss frequencies required to align the modelled ∑PANs at daytime with the observations were $5.5 \times 10^{-4}$ $s^{-1}$ and $2.6 \times 10^{-3}$ $s^{-1}$ for the low and high precursor day, respectively.

The results shown above indicate that daytime physical losses are significantly higher than at nighttime, for which there are several potential explanations. The high daytime values could be explained by a reduction in the surface resistance to foliar uptake when plant-stomata are open, (similar to $O_3$, Shepson et al. (1992)) or a reduction in the transport resistance to uptake due to turbulent mixing. Rapid vertical mixing (venting) out of the canopy would also contribute to the net physical losses during daytime (Bohn, 2006) if significant concentration gradients exist. However, as effects of venting were not observed for alkyl nitrates (i.e. no significant difference between physical loss frequency during day and night was observed), we do not consider venting to be significant.

The overall loss of PAN is caused by a combination of physical losses such as deposition, and (at daytime) transport and thermal decomposition, which are plotted together in Figure S7C and D. The thermal lifetime derived in Figure 7B from the thermal decomposition rate coefficient need to be modified to account for reformation via reaction between $RC(O)O_2$ and $NO_2$. The correction factor ($f_{NO2}$) is described in equation (10) and represents the fraction of $RC(O)O_2$ formed from the thermal decomposition that does not lead to reformation of PANs.

$$f_{NO2} = 1 - \left( \frac{k_{RC(O)O2+NO2}[NO_2]}{k_{RC(O)O2+NO2}[NO_2]+k_{20}[NO]+k_{21}[XO_2]} \right) \qquad (10)$$

where $k_i$ is the rate coefficient for reaction (Ri), and $[NO_2]$, $[NO]$, and $[XO_2]$ are the concentrations of $NO_2$, $NO$, and $XO_2$ ($HO_2+RO_2$). As seen in equation (10), this correction factor requires information not only of $NO$ and $NO_2$, but also $XO_2$ concentrations. In the box model, $NO$ and $NO_2$ are constrained to the measurements whereas $XO_2$ is calculated. The modelled $XO_2$ is compared to the measured value in Figure S8 for the two chosen days. On the
low precursor day, the model and measurements agree well, but on the high precursor day, the model predicts that $XO_2$ should be 4 times higher than the measurements at midday. As the measured $XO_2$ is strictly a lower limit, the effective thermal decomposition rate coefficient for PAN (considering recombination of $CH_3C(O)O_2$ with $NO_2$, see above) was therefore calculated using both the measured and the modelled $XO_2$ to evaluate the uncertainty in this parameter. If
the measured $XO_2$ is correct (and the modelled values are too high on the day during phase 2), the modelled derived physical loss frequency would be too low and would have to be incremented by the difference in effective thermal decomposition frequencies (approximately $5 \times 10^{-4}$ $s^{-1}$). Thermal decomposition accounts for approximately 40 % of the total loss when using modelled $XO_2$ and optimizing for daytime agreement (see Figure S7C and D). The 40%
contribution of thermal loss is inconsistent with the observations of Wolfe et al. (2009) who showed through flux measurements that thermal losses of PAN were dominant. Combining physical and thermal loss rates results in daytime lifetimes of $0.42 \pm 0.05$ h and $0.08 \pm 0.01$ h for the low and high precursor day, respectively. These are significantly shorter than those determined at nighttime (see above or Table 2).

## 5    Conclusion:

Measurements of $NO_x$, $O_3$, BVOCs, the sum of alkyl nitrates ($\sum$ANs), and the sum of peroxycarboxylic nitric anhydrides ($\sum$PANs) have been used to analyse the sources, sinks and lifetime of ANs and PANs in a temperate forest influenced by anthropogenic emissions.

The ANs analysis has been performed for two phases. The first phase is characterised by relatively low temperatures, oxidants, NO and BVOCs compared to the entire campaign, and the second phase was characterised by higher temperatures, oxidants, NO and BVOCs. This led to significantly different production rates, but very similar lifetimes. The production was dominated by OH-initiated reactions at midday for both phases, but large differences were
estimated at nighttime. $NO_3$-initiated reactions play a similarly important role as OH at nighttime for the second phase, however, for the first phase OH still dominates at nighttime. $NO_3$-initiated reactions have also been shown to be important at daytime despite the rapid photolysis. The lifetime for both phases was short at 1-4 hours, which agrees with previous studies in forest environments.

For the PANs analysis, a box model was used to simulate two individual days; one in the first phase of the ANs analysis and one in the second phase. Two constant physical loss terms are applied for each of the two days optimized to match the average daytime and nighttime mixing ratios. For the low precursor day (July 4th 2022), loss frequencies of $5.5 \times 10^{-4}$ $s^{-1}$ and $1.8 \times 10^{-4}$ $s^{-1}$ were used to align measurement and model for daytime and nighttime, respectively, while
for the high precursor day (July 13th 2022) $2.6 \times 10^{-3}$ $s^{-1}$ and $1.1 \times 10^{-3}$ $s^{-1}$ were used. This resulted in lifetimes of around 20 min and 4 min at midday for the low and high precursor day,

respectively, where thermal decomposition contributed approximately 40 %. Peroxyacetic nitric anhydride (PAN) represents 48-78% of ∑PANs according to the box model, with the highest fractions predicted at daytime.

Lifetimes of organic nitrates in the forested environment are very short. A potential reason for the short lifetime is dry deposition to e.g. soil and foliar surfaces. If deposition is the cause of the short lifetimes this would be consistent with a picture of the forest ecosystem capturing essential nitrogen-containing nutrients originating from anthropogenic sources and transferring them to the biosphere.

## 6    Data Availability:

All measurements from the ACROSS campaign including $NO_x$ (Andersen and Crowley, 2023a; Xue et al., 2023), $O_3$ (Crowley, 2023), organic nitrates (Andersen and Crowley, 2023b), $NO_3$ reactivity (Dewald and Crowley, 2023),  BVOCs (Michoud et al., 2024), meteorological

quantities (Denjean, 2023), photolysis frequencies (Dusanter and Jamar, 2023), OH (Kukui, 2023a), and peroxy radicals (Kukui, 2023b) can be found at https://across.aeris-data.fr/catalogue/ (last access: 31 August 2024).

## 7    Author contribution:

All authors contributed with measurements. Data analysis was conducted by STA with contributions from JNC. RS did the box modelling. CC and VM organized the field campaign with contributions from the individual group leads. STA and JNC developed the manuscript with contributions from all authors.

## 8    Competing Interests:
The authors declare that they have no conflict of interest.

## 9    Acknowledgements:
STA is thankful to the Alexander von Humboldt foundation for funding her stay at MPIC.

PD gratefully acknowledges the Deutsche Forschungsgemeinschaft (project "MONOTONS", project number: 522970430).

We thank Sebastien Dusanter and Marina Jamar from IMT Nord Europe, France, for supplying photolysis frequency measurements on the tower.

The ACROSS project has received funding from the French National Research Agency (ANR)
under the investment program integrated into France 2030, with the reference ANR-17-MPGA-0002, and it was supported by the French National program LEFE (Les Enveloppes Fluides et l'Environnement) of the CNRS/INSU (Centre National de la Recherche Scientifique/Institut National des Sciences de l'Univers). CNRS-INSU provides support to the PEGASUS platform as a national facility. Data from the ACROSS campaign and the PEGASUS facility are hosted
by the French national center for Atmospheric data and services AERIS.

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

     **11  Figures:**

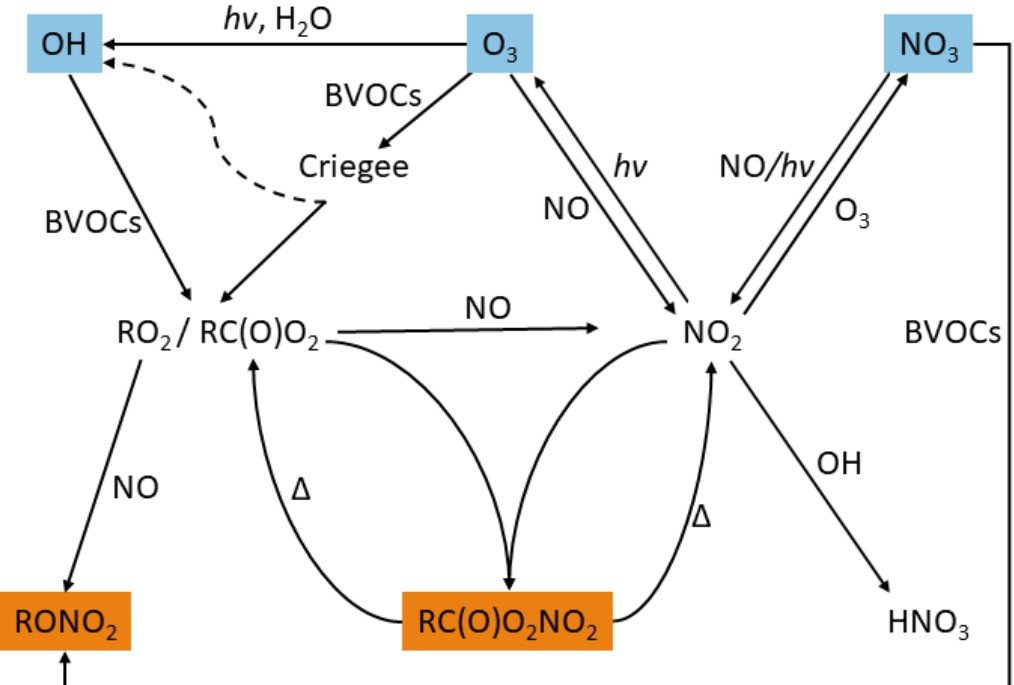

Figure 1: Schematic diagram showing the formation of PANs ($RC(O)O_2NO_2$) and ANs ($RONO_2$)
from the oxidation of VOCs by OH, $O_3$, and $NO_3$. Reactions of $RO_2$ not relevant to the formation
of $RC(O)O_2NO_2$ and $RONO_2$ have been left out.  Note that the scheme does not attempt to capture
all formation routes of the primary oxidants, especially those of the OH-radical, which may
additionally be formed in e.g. reactions of $HO_2$ with NO and photolysis of HONO.

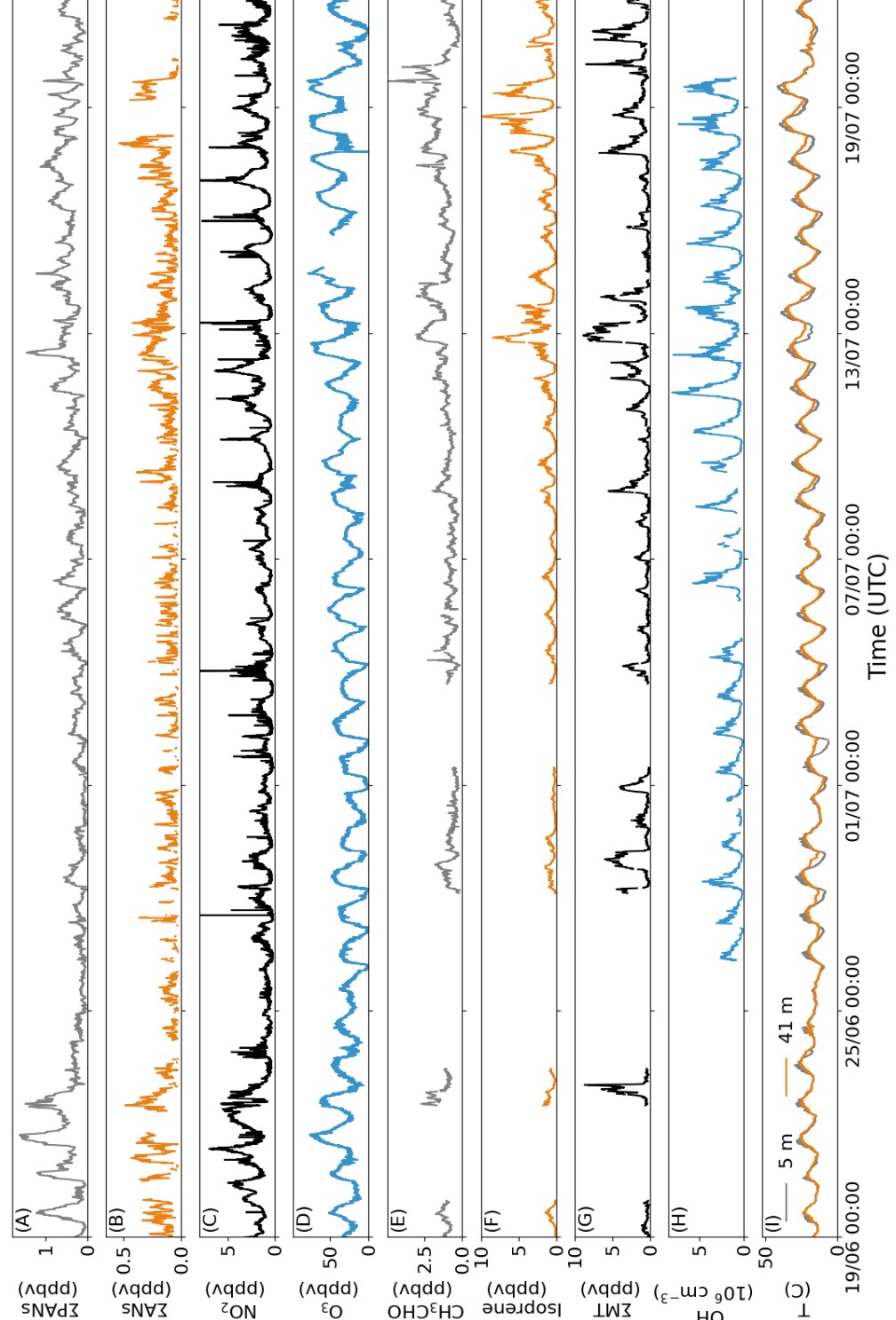

Figure 2: Time series of 10-minute averages of $\sum$PANs, $\sum$ANs, acetaldehyde, isoprene, and sum
of monoterpenes ($\sum$MT), 12-minute averages of OH radicals, and 1-minute averages of $NO_2$, $O_3$,
and temperature at 5 and 41 m during the ACROSS campaign.

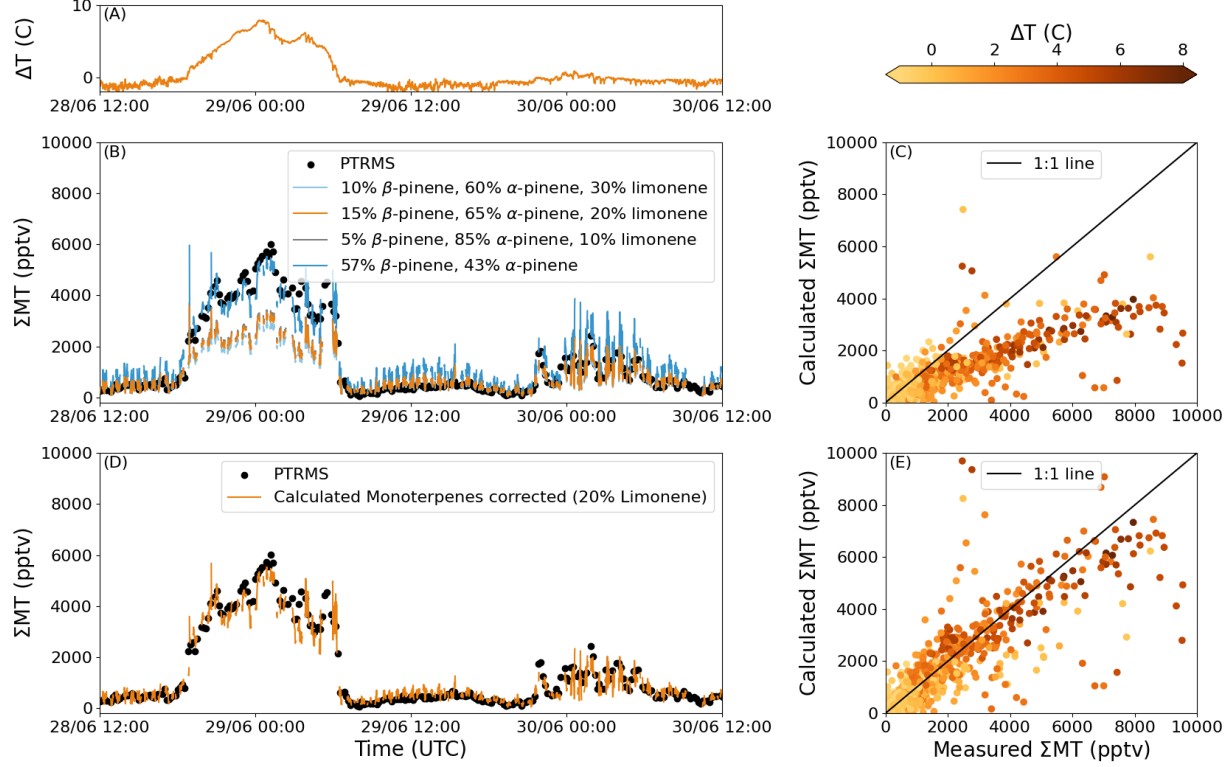

Figure 3: NO$_3$-reactivity based estimation of monoterpene mixtures during ACROSS. Panel A shows the difference in temperature between 41 m and 5 m ($\Delta T = T_{41m}-T_{5m}$) for 48 hours during ACROSS. Panel B shows the measured sum of monoterpenes ($\sum$MT) and that calculated using eq. (1) with four different monoterpene mixtures. Panel C plots calculated against measured $\sum$MT for the scenario with 20% limonene coloured by $\Delta T$. Panel D shows the scenario with 20% limonene from panel C using the mixture of 57% β-pinene and 43% α-pinene when $\Delta T>1°C$ and panel E shows the scatter plot after performing the correction for temperature inversions.

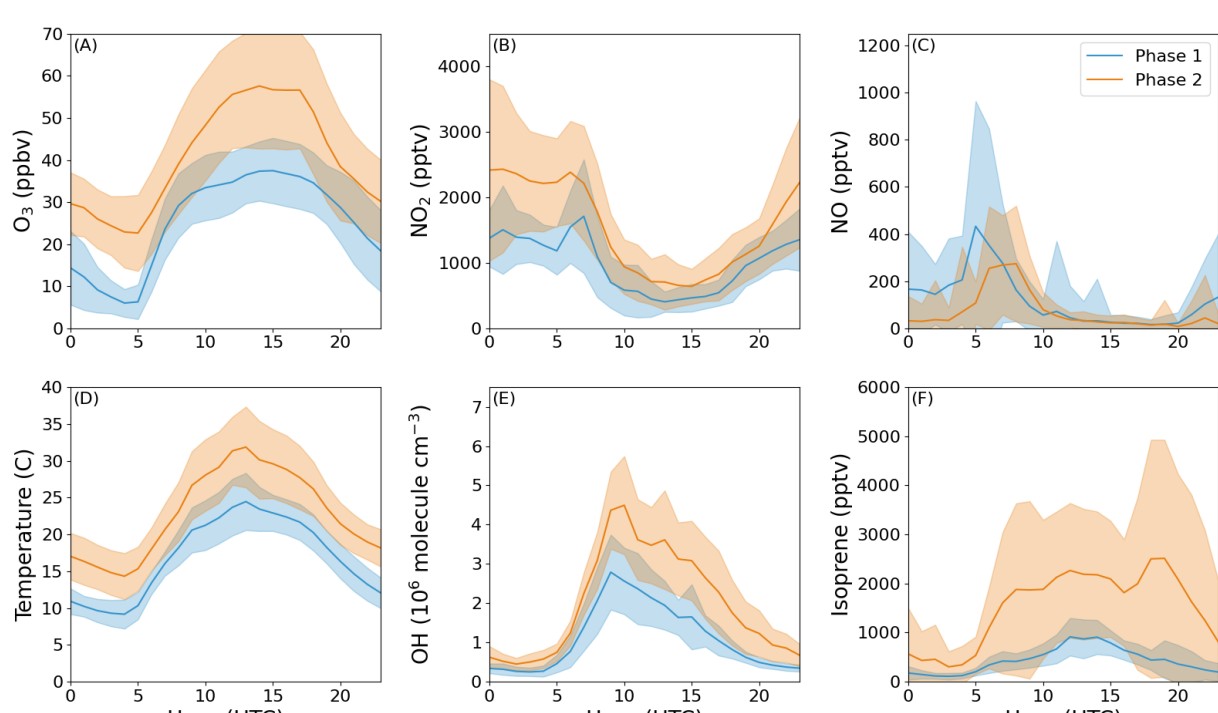

Figure 4: Diel profiles of $O_3$, $NO_2$, NO, temperature, OH, and isoprene for the two phases used to analyse organic nitrates during the ACROSS campaign; phase 1 from June 28[th] 2022 to July 7[th] 2022 and phase 2 from July 8[th] 2022 to July 20[th] 2022. The shaded area is ±1σ.

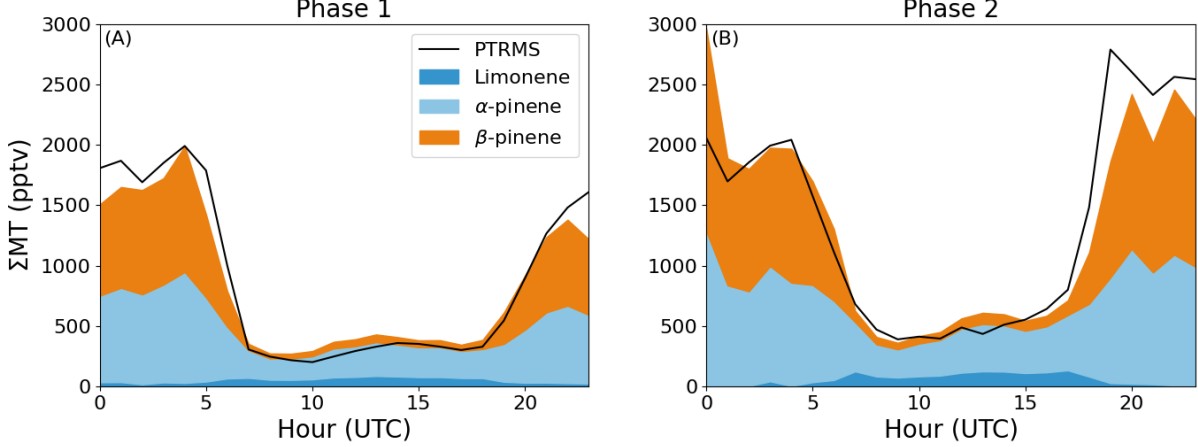

Figure 5: Average diel profiles of the PTRMS measurements of the sum of monoterpenes (black)
and the calculated, average mixture of limonene (dark blue), α-pinene (light blue), and β-pinene
(orange) for the scenario with 20% limonene.

943

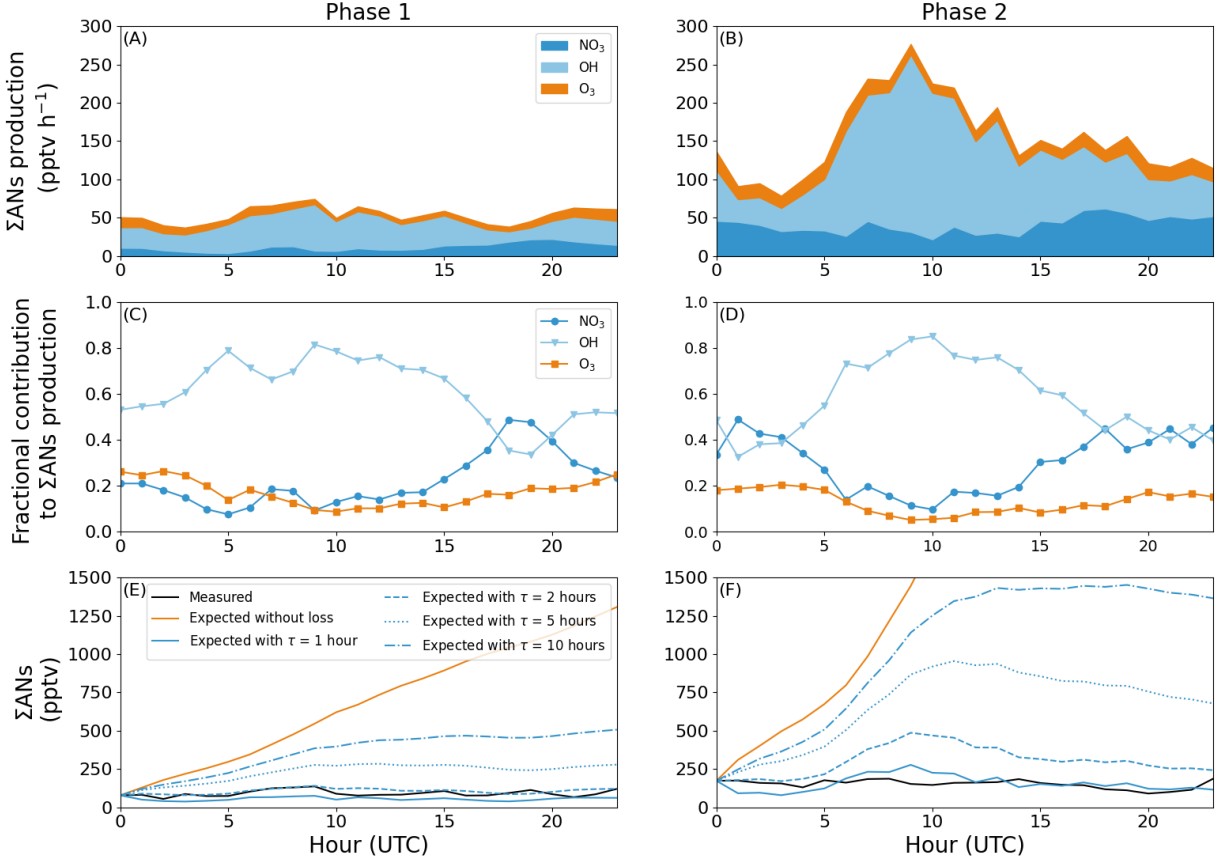

944

Figure 6: Panel A and B show the mean diel profiles of the $\sum$ANs production rates from $NO_3$-, OH-, and $O_3$-initiated oxidation of a monoterpene mixture consisting of 20% limonene, 15% β-pinene, and 65% α-pinene for phase 1 and 2, respectively. The fractional contribution to the $\sum$ANs for each oxidant is plotted in panel C and D. Panel E and F show the average diel profile across the two phases for the measured alkyl nitrates (black), expected alkyl nitrates with (blue) and without (orange) any losses, where the different blue lines were calculated using different effective lifetimes of the alkyl nitrates.

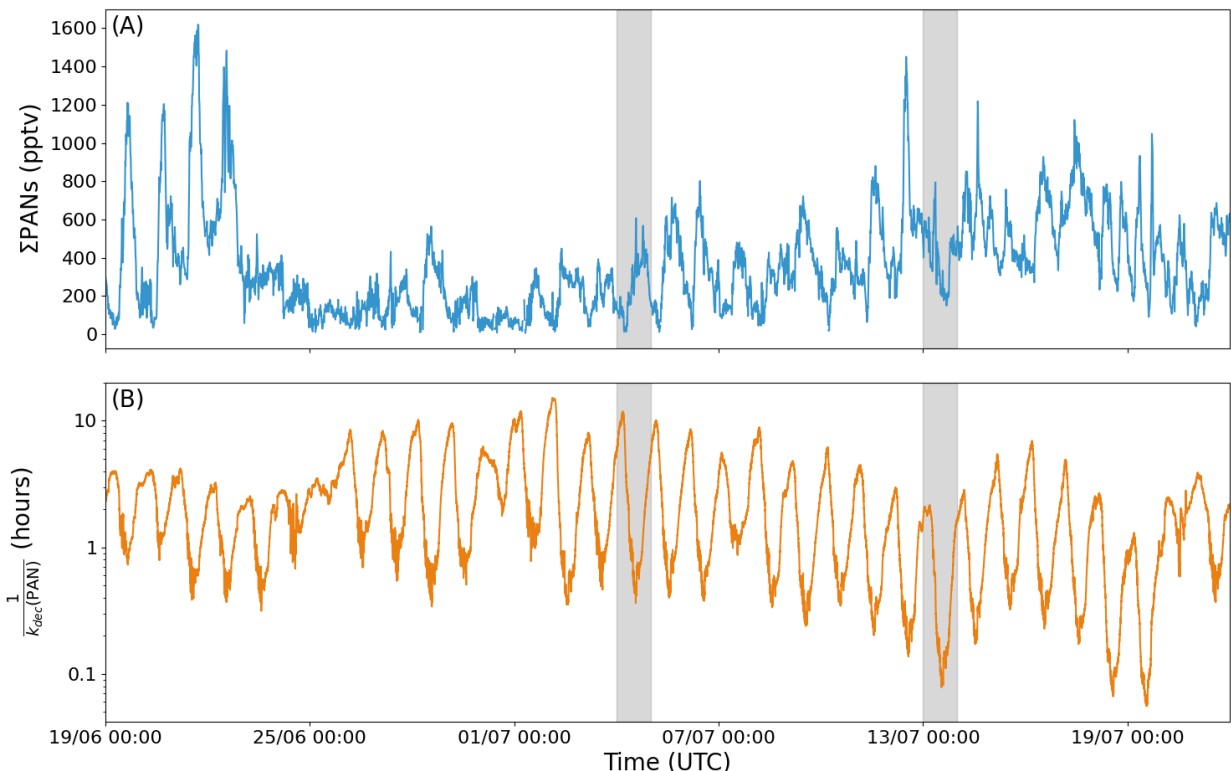

952

Figure 7: Time series of ∑PANs (A) and the thermal lifetime of PAN (B) during ACROSS, where $k_{dec}$ is the temperature dependent rate coefficient for the thermal decomposition of PAN using the expression preferred by the IUPAC panel (IUPAC, 2024). Days marked in grey are used for analysis in Figure 8.

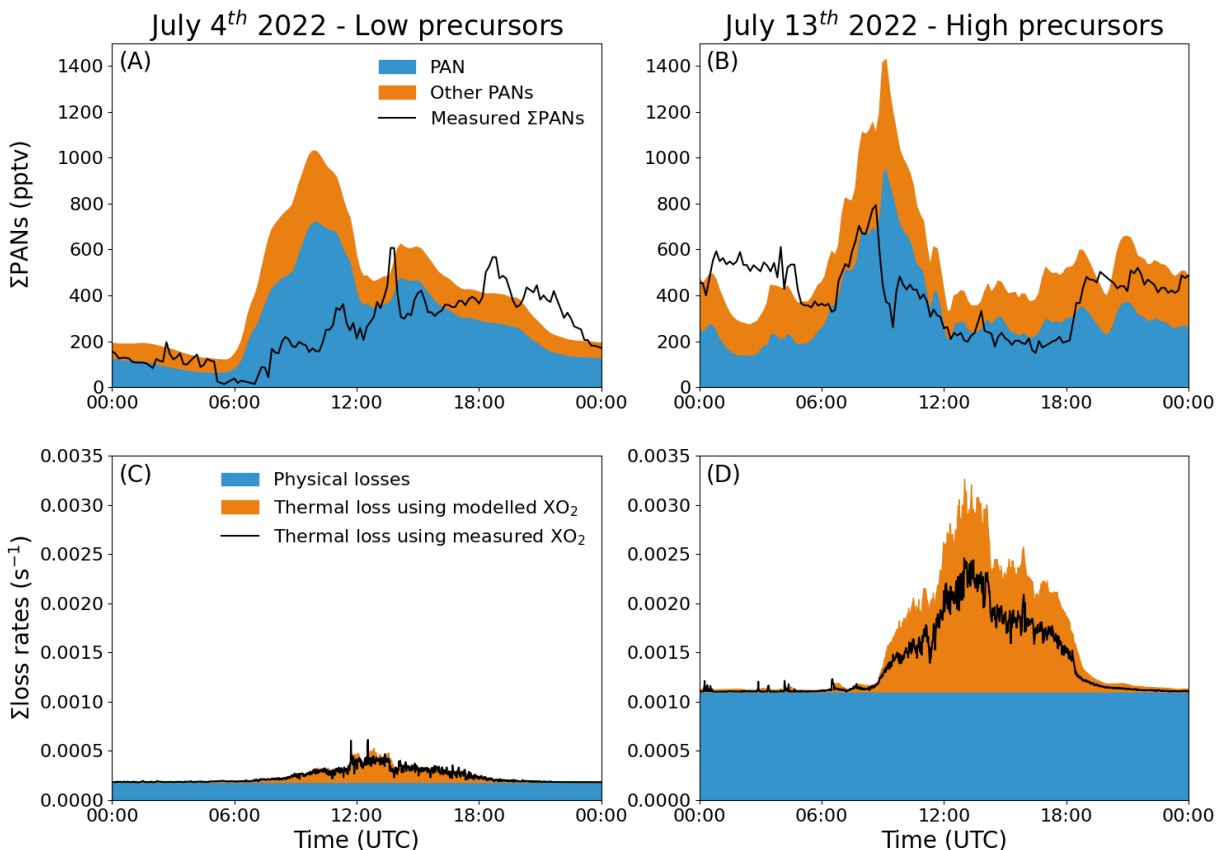

Figure 8: The measured and modelled when optimizing for nighttime agreement $\sum$PANs is plotted for two individual days; one with low precursors (A) and one with high precursors (B). The optimised physical loss for each day is shown in panel C and D together with the thermal decomposition when taking recombination into account using both the measured and modelled mixing ratio of $XO_2$.

## 12 Tables:

**Table 1: Rate coefficients and relevant yields for the calculation of $\sum P_{ANs}$**

| VOC | $k(NO_3)$ at 298 K (cm$^3$ molecules$^{-1}$ s$^{-1}$) | $\alpha^{NO_3}$ | $k(OH)$ at 298 K (cm$^3$ molecules$^{-1}$ s$^{-1}$) | $\alpha^{RO_2+NO}$ | $k(O_3)$ at 298 K (cm$^3$ molecules$^{-1}$ s$^{-1}$) | $\alpha^{O_3}$ [i] |
|---|---|---|---|---|---|---|
| α-pinene | $6.2 \times 10^{-12}$ (± 26%) [a] | 0.18 [b] | $5.3 \times 10^{-11}$ (± 20%) [a] | 0.22 [f] | $9.6 \times 10^{-17}$ (± 41%) [a] | 0.80 [a] |
| β-pinene | $2.5 \times 10^{-12}$ (± 32%) [a] | 0.49 [c] | $7.6 \times 10^{-11}$ (± 12%) [a] | 0.24 [g] | $1.9 \times 10^{-17}$ (± 78%) [a] | 0.30 [a] |
| $d$-limonene | $1.2 \times 10^{-11}$ (± 32%) [a] | 0.50 [d] | $1.7 \times 10^{-10}$ (± 12%) [a] | 0.23 [g] | $2.2 \times 10^{-16}$ (± 26%) [a] | 0.66 [a] |
| Isoprene | $6.5 \times 10^{-13}$ (± 41%) [a] | 0.77 [e] | $1.0 \times 10^{-10}$ (± 15%) [a] | 0.13 [h] | $1.28 \times 10^{-17}$ (± 20%) [a] | 0.26 [a] |

$\alpha^{NO_3}$: Yield of ANs from NO$_3$+BVOC in air.

$\alpha^{RO_2+NO}$: Yield of ANs from RO$_2$+NO for the specific BVOC when the RO$_2$ is formed from BVOC+OH.

$\alpha^{O_3}$: Yield of RO$_2$ from the ozonolysis of BVOC in air.

[a] Rate coefficients and yields recommended by IUPAC(IUPAC, 2024; Mellouki et al., 2021; Cox et al., 2020).

[b] Average of Wängberg et al. (1997), Berndt and Böge (1997), Hallquist et al. (1999), Spittler et al. (2006), Fry et al. (2014), and Devault et al. (2022).

[c] Average of Hallquist et al. (1999), Fry et al. (2009), Fry et al. (2014),Claflin and Ziemann (2018), and Devault et al. (2022).

[d] Average of Hallquist et al. (1999), Spittler et al. (2006), Fry et al. (2011), Fry et al. (2014), and Devault et al. (2022).

[e] Average of Barnes et al. (1990), Berndt and Boge (1997), Perring et al. (2009), Kwan et al. (2012), Rollins et al. (2009), and Schwantes et al. (2015).

[f] Average of the yields given by Nozière et al. (1999) and Rindelaub et al. (2015).

[g] Perring et al. (2013).

[h] Recommended by Wennberg et al. (2018) based on multiple studies.

[i] Set equal to the OH yield of the ozonolysis since a RO$_2$ is formed with each OH.

**Table 2: Overview of the thermal, physical and total loss frequencies (and lifetimes) of ANs and PANs needed to explain the measured ANs and PANs during ACROSS.**

| | | Phase 1 (Low photochemical activity) | | Phase 2 (High photochemical activity) | |
|---|---|---|---|---|---|
| | | Day | Night | Day | Night |
| ANs | Thermal loss frequency (s$^{-1}$)[a] | - | - | - | - |
| | Physical loss frequency ($10^{-4}$ s$^{-1}$) | 1.7 ± 0.3 | 1.7 ± 0.4 | 3.3 ± 0.8 | 2.5 ± 0.9 |
| | Total loss frequency ($10^{-4}$ s$^{-1}$) | 1.7 ± 0.3 | 1.7 ± 0.4 | 3.3 ± 0.8 | 2.5 ± 0.9 |
| | Total lifetime (h) | 1.7 ± 0.2 | 1.7 ± 0.5 | 0.9 ± 0.2 | 1.3 ± 0.5 |
| PANs | Thermal loss frequency ($10^{-4}$ s$^{-1}$)[a] | 1.2 ± 0.8 | 0.1 ± 0.1 | 8.8 ± 5.9 | 0.8 ± 1.3 |
| | Physical loss frequency ($10^{-4}$ s$^{-1}$)[b] | 5.5 | 1.8 | 26 | 11 |
| | Total loss frequency ($10^{-4}$ s$^{-1}$) | 6.7 ± 0.8 | 1.9 ± 0.1 | 34.8 ± 5.9 | 11.8 ± 1.3 |
| | Total lifetime (h) | 0.42 ± 0.05 | 1.5 ± 0.1 | 0.08 ± 0.01 | 0.24 ± 0.02 |

Day is defined as 06:00-18:00 UTC (08:00-20:00 LT). Night is defined as 18:00-06:00 UTC (20:00-08:00 LT).

All values except the modelled physical loss frequencies for PANs are given as the mean ± 1σ. The ANs are based on hourly averages for each phase and the PANs are based on modelled loss frequencies for one day during each phase.

[a] The thermal loss frequency has been corrected for the fractional recombination of CH$_3$C(O)O$_2$ with NO$_2$.

[b] In total four model runs were conducted to examine the two phases and to differentiate between day and night.