# Peer review of "Short lived organic nitrates in a sub-urban temperate forest: An"

_EGUsphere, 2024_

## Referee Comment (RC3)

Review of "Short lifetimes of organic nitrates in a sub-urban temperate forest indicate efficient assimilation of reactive nitrogen by the biosphere," Andersen et al., EGUsphere (2024)

9 December 2024

**Summary**

This paper presents ground-based observations of trace gases in a temperate forest. They use a combination of simple models to calculate production rates of total alkyl nitrates and peroxyacyl nitrates, and they examine how these depend on different oxidants and how they vary between different periods. They use the residual between modeled and observed mixing ratios to infer physical loss rates, which they imply reflect loss of these compounds via dry deposition. The writing is good and the number and style of figures is appropriate.

As detailed below, I find several major flaws in this study, specifically with regards to 1) presumption of the nature of missing sinks, 2) inadequate treatment of uncertainty, and 3) missing chemical reactions and inadequate consideration of the applicability and limitations of the model for this particular environment. Any one of these would be cause for major revisions, but in sum I regret that they justify rejection (with encouragement to resubmit).

**General Comments**

The title is misleading. The results presented in this paper do not "indicate efficient assimilation of reactive nitrogen by the biosphere." At best, results indicate that additional sinks are sometimes needed to square a steady state model with observed concentrations. No evidence is presented that can isolate these sinks. Indeed, L554 (last paragraph of the conclusions), states that deposition is "presumably" the cause. The title should be amended to reflect defensible conclusions.

Missing sinks: equating model-estimated "physical losses" with deposition is quite a leap. First, a steady-state 0-D box model cannot be expected to represent all processes accurately, or even sufficiently, within a forest where there are complex interactions of emissions, radiation, and chemistry. Numerous studies using 1-D canopy models support this. The transport timescale in a typical forest canopy is a few minutes, so anything with a lifetime longer than this would be affected by those processes. Second, it is possible to estimate the expected dry deposition rates using a range of reasonable deposition velocities (Nguyen et al., 2015, and other studies). For a 1 km boundary layer and Vd = 1 – 3 cm/s, the effective deposition lifetime is 9 – 27 h, which is much longer than the 1.5 h AN lifetime estimated here. Several studies have shown that aerosol formation is important (L69). OK, that's the rant; the point is that the analysis could be improved by 1) estimating deposition loss rates based on literature, and 2) not presuming that "physical loss" means deposition.

Nighttime OH: L261 states that nighttime OH can be explained by ozonolysis, but analysis is never shown to support this. Later, L346 states that a major difference between this study and a previous

one is the nighttime OH data. This is not the first time nocturnal OH has been observed, and I suspect the jury is still out on the extent to which this might be a measurement artifact. Given this appears to be important, the paper should include some analysis of whether or not the observed nighttime OH is consistent with known chemistry.

Uncertainty: Measurement uncertainties are not propagated. This is especially important for any calculations using OH. Figures 6 C-D could use error bars or something in the text discussing how uncertain that balance is. Figure 8 A-B would also benefit from an uncertainty estimate on the modeled PAN – at least from known errors. This doesn't have to be fully rigorous to be useful.

**Specific Comments**

Sect. 3.2.1: What is the accuracy of the reactive N measurements?

L124: Corrected for what?

Sect. 3.2.2: how confident are you that the observed OH is representative of the average for the forest canopy or the larger environment that would contribute to VOC oxidation? If you are in a forest, presumably there is significant shading.

Sections 4.1 – 4.3 present theory that is used to analyze the results, not the results themselves. As such, this should be moved to Methods or a "Theory" section.

Sect. 4.1 does not include $N_2O_5$. This seems like a major omission, especially considering Sect. 3.2.1 states $N_2O_5$ was measured. This needs to be included in the $NO_3$ steady state calculation (including heterogeneous loss of $N_2O_5$), or a quantitative argument needs to be made on why it can be excluded.

L280: What is the basis for the generic rate coefficients?

L376: This is also consistent with another very different analysis of isoprene nitrates, which found lifetime of < 2h with losses dominated by aerosol uptake (Wolfe et al, 2015, Table S5).

L378: a potential implication of $XO_2$ over-prediction is that OH is too high. How would reduced OH alter your estimated AN lifetime? At a minimum, this should be considered from the standpoint of the presumed 25% uncertainty in observed OH.

L405: MPA radical can also isomerize (Crounse et al., 2012). Presumably this has a relatively minor impact on modeled PANs.

L465: This is also consistent with analysis of PAN fluxes above another forest (Wolfe et al., 2009), although in that study they found a downward flux driven by warmer temperatures at the surface (meaning that canopy transport was a net source of PAN).

**Technical Comments**

L29: Define "ACROSS"

L53: capitalize IUPAC

L146: suggest striking this sentence. I believe it is no longer acceptable to refer to an unwritten paper, especially since they often remain unwritten.

L160: define LISA

L230" delete "i.e. . . . terms."

L269: OH addition, which is the dominant channel for alkenes, does not lead to $H_2O$ formation.

L378: change "above" to "Sect. 3.3."

L498: Fig. 8

**References**

Crounse, J. D., et al.: On the atmospheric fate of methacrolein: 1. Peroxy radical isomerization following addition of OH and O2, Journal Of Physical Chemistry A, 116, 5756–5762, https://doi.org/10.1021/jp211560u, 2012.

Nguyen, T. B., Crounse, J. D., Teng, A. P., St. Clair, J. M., Paulot, F., Wolfe, G. M., and Wennberg, P. O.: Rapid deposition of oxidized biogenic compounds to a temperate forest, Proceedings of the National Academy of Sciences, 112, E392–E401, https://doi.org/10.1073/pnas.1418702112, 2015.

Wolfe, G. M., et al.: Eddy covariance fluxes of acyl peroxy nitrates (PAN, PPN and MPAN) above a Ponderosa pine forest, Atmospheric Chemistry And Physics, 9, 615–634, 2009.

Wolfe, G. M., et al.: Quantifying sources and sinks of reactive gases in the lower atmosphere using airborne flux observations, Geophysical Research Letters, 42, 8231–8240, https://doi.org/10.1002/2015GL065839, 2015.

---

## Author Response (AR1)

We thank both reviewers for their thorough readthrough of the manuscript and for their comments. Below we have written the comments in black and our replies in red.

Reviewer comment 1:

This paper reports new measurements of organic nitrate concentrations and reactivity, alongside box modelling, in order to constrain the lifetime of alkyl and peroxyalkyl nitrates. The analysis uses measurements made during the ACROSS campaign in 2022, at a tower sit in a Rambouillet forest, 50 km southwest of Paris, France. The major result is an apparently very short (deposition) lifetime for these organic nitrate species over the forest, due to their low observed concentrations. The modelling applied to interpret the data is rather complex. Because no speciated monoterpene data was available, different MT mixtures were tested, including a cutoff correction to treat the effect of (frequent) thermal inversion periods on the MT mixture. The complexity of the modelling made the analysis a bit difficult to follow and given the goal of determining lifetimes, I wonder if it could be made simpler / clearer.

Our derivation of lifetime requires calculation of production rates of both ANs and PANs. This was done using an analytical expression for ANs, but required box modelling for PANs. In the absence of measurements of speciated monoterpenes we had to make some assumptions, which necessitated a sensitivity test of this assumption (i.e. considering different mixtures of MT). This is an important aspect of the uncertainty estimation and unavoidable.

Of larger concern are the surprisingly high inferred contribution of OH to nitrate formation at night (when OH contributions are typically considered to be low), and the 4 x model overestimation of XO2 concentration relative to measurements. Could there be biases in these OH and XO2 measurements that skew this analysis substantially?

The contribution of OH to nighttime nitrate formation is based on measurements of OH with an uncertainty of +-25%. The source of OH at nighttime (ozonolysis of olefins) is well established. We are aware that some previous measurements of OH in forested regions were biased by interfering "background" signals related to ozonolysis of unsaturated VOCs. However, the measurement technique used here accounted for the production of OH in the instrument (see specific comment below) and we are confident that the OH concentrations that we report are free of this systematic bias. As not all $RO_2$ are detected with the same sensitivity by the instrument, the XO2 is strictly speaking a lower limit (see specific comment below).

The results of this paper, showing rapid deposition of organic nitrates to a forest, are quite interesting and will be of strong interest to the nitrogen cycle community. I hope the authors can respond to the below comments and questions to help make the report clearer and clarify the limitations of the interpretation.

The short lifetimes of the organic nitrates certainly have repercussions for the nitrogen cycle in forested regions. We have taken the reviewers comments on board and improved the clarity of the interpretation accordingly.

General Comments:

1) You use the term "peroxycarboxylic nitric anhydrides" throughout for PAN. I understand this is a correct IUPAC name, but I think "peroxyacetyl nitrate" is more commonly used in atmospheric chemistry community – might make your paper clearer understood (and more likely successfully searched for) if you use the latter name?

We have added a sentence the first time PANs are mentioned in the introduction to describe that PAN is included in PANs and to mention the more common name for PAN in the abstract: "(PAN, commonly known as peroxyacetyl nitrate)"

2) Section 3.1: You refer to the measurement site as a tower in a clearing, which to me sounds like no trees directly around the tower, but later it's clear that your inlets are partly shaded (you mention the difference in photolysis rates if you were to use the above-clearing radiation measurements). Were they in the woods? Or are the trees so tall that even faraway trees shade your inlets for substantial parts of the day? I think a photo or diagram of the site might help here. And sometime when you talk about inside or outside the clearing, do you mean on the tower above the treetops? Some clarifying language could help here.

A detailed description of the site has been added to the SI as figure S1, which includes a schematic showing the approximate positions of all the containers, the dimensions of the site (also added in the text) and a photo from the tower looking down on some of the containers. The photo shows how close the trees were and the shading of the MPIC container at 16:45 local time.

Inside the clearing refers to the measurements made at 3-5 m, above the clearing refers to tower measurements and outside the clearing refers to the natural forest. This has been made clearer in the text.

3) in line109 you introduce the variable kNO3, and in line 117 you say you will henceforth refer to this as kBVOC because BVOC losses are dominant. But later you introduce additional k's to describe the losses to various oxidants, so it's confusing to lose the oxidant here. Why not kNO3+BVOC, to be completely clear?

kNO3 has been removed from line 109, so only kBVOC is mentioned in this paragraph and in section 3.2.4 it has been made clear what is meant with the different k's.

4) around line 120: was O3 measured on the same inlet line? (the diagram of relevant sampling instrumentation would avoid readers wondering things like this)

As mentioned at the beginning of section 3.2.1 (line 90-95), O3 was measured using co-located inlets inside the MPIC container. A diagram of the different containers has been added to the SI.

5) Because your OH and XO2 results are so surprising, I think the descriptions of the OH and XO2 measurements in section 3.2.2. are key. Can you add some discussion of potential interferences or reasons for non-zero backgrounds? And explain how the XO2 is determined subtractively from the NO converted channel minus no NO channel, and what the associated uncertainties are. Is there any possibility of additional reactions between the conversion reactor and measurement? (diagram helpful here too)

A detailed description of the CIMS instrument used here for the XO2 and OH measurements has been presented previously (Kukui et al., 2008; Kukui et al., 2021) and also will be given in forthcoming publication about OH and XO2 chemistry during the ACROSS campaign. In brief, the background signal is determined by adding $NO_2$ as OH and XO2 scavenger before their conversion to H2SO4. As discussed in Kukui et al., 2021, formation of H2SO4 in the reaction of stabilized Criegee intermediates (SCI) with SO2 in the conversion reactor may lead to some positive interference in radical measurements. However, this interference (several $10^4$ radicals $cm^{-3}$) is negligible compared to the measured OH concentrations. The latest configuration of the chemical conversion and ion molecule reactors is presented in Figure 1S of Kukui et al., 2021.

Text has been added to the manuscript to explain potential interferences.

Kukui, A., Ancellet, G., and Le Bras, G.: Chemical ionisation mass spectrometer for measurements of OH and Peroxy radical concentrations in moderately polluted atmospheres, Journal of Atmospheric Chemistry, 61, 133-154, 10.1007/s10874-009-9130-9, 2008.

Kukui, A., Chartier, M., Wang, J., Chen, H., Dusanter, S., Sauvage, S., Michoud, V., Locoge, N., Gros, V., Bourrianne, T., Sellegri, K., and Pichon, J. M.: Role of Criegee intermediates in the formation of sulfuric acid at a Mediterranean (Cape Corsica) site under influence of biogenic emissions, Atmos. Chem. Phys., 21, 13333-13351, 10.5194/acp-21-13333-2021, 2021.

6) around line 153, where you talk about the neglect of upwelling radiation which could cause an underestimate of 5-10%, it would be helpful to know a bit more what the canopy is like. (again, perhaps a diagram + photo figure would help here). How shaded is the container? Maybe add a plot comparing the top of tower radiation to that at the measurement container, and show the percentage difference over the daily cycle.

A plot (S3) has been added to the supplementary information showing the differences in JNO3 between the measurements on top of the tower and on top of the MPIC container and text has been added to the instrumental section. The shading of the MPIC container can also be observed on the photo in the new Figure S1.

7) around line 165: At what height was the PTR-MS sampling? (again, diagram would help). Maybe discuss this in the context of your MT composition estimates – are you measuring at the same position as the reactivity or a different height, and might there be gradients?

The PTRMS measurements were conducted at 4.6 m, which is at a similar height to that of the reactivity, however, the inlets were not co-located and there could therefore be horizontal gradients that we neglect in our analysis. Text has been added to discuss this.

8) around line 185: the limonene fraction sensitivity analysis is rather confusing. It's not clear why the specific 3 mixtures are chosen, and in Figure 3B it doesn't look like the fraction of limonene matters much. What matters is that most of the time, you actually use the delta-T cutoff and assume you don't have any limonene at all in the reactivity mix.  What motivated the choice of delta-T = 1 for the cutoff? This feels strangely arbitrary, and in figure 3 you see that the cutoff case seems to apply to most of the data. Why not just omit limonene altogether and make your discussion less complicated?  Or at least, if the limonene does affect the low concentrations / daytime modelling, why present the 3 scenarios and not just your best estimate of the daytime vs. limonene fraction? This brings a lot of extra discussion that may confuse your main message.

We have now emphasized that these mixtures are examples of potential mixtures that reproduce both the observed $NO_3$-reactivity and the sum of monoterpenes measured by the PTRMS. We find that with some constraint (based in rate coefficients and yields) we can mix the three monoterpenes in different ways and show that while some impact is seen, the final mixture does not strongly affect the calculated lifetimes or alter the conclusions of the manuscript.

However, the mixture with 57% beta-pinene and 43% alpha-pinene has been added to Figure 3B to show why we cannot use this mixture for the entire time period – it significantly overestimates the calculated sum of monoterpenes when no temperature inversion is observed. Text has been added to the section to clarify how the mixtures have been calculated and that we use the mixture without limonene (only) 26% of the time.

9) towards the end of section 4.2: How did you partition XO2 into RO2 and HO2 to determine how to weight the RO2 / HO2 rate constants? Or you could just show the reaction as kXO2 and explain how you estimated that aggregated rate constant.

We have chosen to mention both reactions (i.e. with $RO_2$ and $HO_2$), but have now made it clear with additional text that we use just one "generic" rate coefficient and the measured XO2 in the calculation. We have additionally added text to describe where the generic rate coefficients come from.

10) In section 4.4 where you mention mixture 2 again, could just state the assumed mixture to avoid confusion (why 2?), and if you keep the delta-T threshold, remind the reader of how that works briefly here –e.g., "temperature inversions and resulting assumed removal of all limonene at the surface."  Also, somewhere early in this section say that you report all errors as +/- 1 sigma, and then don't repeat it after every number. At line 333, you mention mixture 2 again – state that at the outset somewhere that it applies to all further analysis, so you don't need to keep interrupting the text with it. Readability gets compromised by these repetitions, and mixture 2 could get confused with phase 2, etc.

The text has been modified so no mixture numbers are mentioned and a reminder of the temperature inversions has been added. And the +/- sigma has been changed to intervals to make it clearer.

11) The very high simulated OH contribution to PANs at night is quite surprising, so it would be good to see how else you can corroborate this. Does your box model reproduce the measured high nighttime OH concentration, with a reasonable assumption of OH recycling from the O3 reactions? (I assume that has to be where it's coming from). Or could there have been some other (spurious?) local source of OH?

We have not examined the sensitivity of our results to changes in the OH concentration as our modelled PAN production term was constrained by observed OH concentrations. The research group that operated the OH instrument is modelling the HOx observed during the campaign and this will be discussed and published elsewhere. As will be shown in a forthcoming publication, the nighttime OH levels are broadly consistent with measured OH-reactivity, HO2-recycling via reaction of HO2 with NO and production via the ozonolysis of terpenoids (this has been added to the text in section 4.2).

12) Lines 378 – 384: the 4 times higher modeled XO2 than measured is also shocking. It doesn't seem satisfactory to just say the origin is not known and then use the higher modeled value. Can you look more into what the highest individual XO2 concentrations are in your model and try to figure out whether there could be some erroneous pathways or some that might not be relevant to your site?

We did not simply use the modelled XO2 concentrations but also the measured XO2 in a sensitivity study. The research group responsible for the XO2 measurements is conducting independent modelling where they also take into account the change in sensitivity of the instrument to different $RO_2$. They observe higher XO2 in the model compared to observations, but less than a factor two. For this reason, using both the measured values (essentially a lower limit) and the modelled values allows us to estimate the uncertainty in our derived lifetimes arising from the reactions of $RO_2$ (which is now treated in greater detail including tabulated uncertainties and their sources in the supplementary information).

13) Lines 480-483: Why do you bring up HPAN here and not when you introduce all the other reactions? Also, the use of an "arbitrary loss term" is not well justified. Why not just omit HPAN formation from the model, if they have never been observed and you have to make them go away with an arbitrarily high loss term to get things right?

HPAN has been observed in experimental studies on the oxidation of glyoxal (HOCH2CHO), where it was also shown to be short lived due to surface losses. The molecule is in the MCM because it is truly generated in the atmosphere. It makes more sense to add a realistic short lifetime to the model to match observations than to change the model by removing its production, which is real.

14) Mid-page 13:  You describe the differences in daytime vs. nighttime physical loss rates based on vertical mixing, but it sounds (here) like in your model you use one physical loss rate. At this point I'm wondering, why not account for the different daytime and nighttime loss rates by tuning them separately? But then on the top of p. 14 you describe separate lifetimes for daytime and nighttime. I'm a bit confused whether these refer to separately tuned parameters in a single model, or the optimized (full-diurnal-cycle)

parameter in cases where you tune to match daytime or nighttime better. I suggest a careful edit of the last 3 paragraphs of this section (lines 484-527) to sharpen and clarify the description of what you did to obtain the day and night lifetime estimates. The table is a helpful addition, but the text is still a bit difficult.

On mid-page 12 (original manuscript), we describe two different physical loss rates tuned to day- and nighttime observations. We have done two separate model runs, where the physical loss rate was fixed to either match day- or nighttime observations. The model run matching the nighttime observations is shown in Figure 8 and the model run matching the daytime observations is shown in the SI. We have clarified the text describing the modelling approach and now write "The thermal losses of PANs were included in the model as described above and an additional first-order loss term (physical losses from e.g. deposition) was varied in two separate model runs to get satisfactory model-measurement agreement for ∑PANs during daytime (06:00-18:00 UTC) and nighttime (18:00-06:00 UTC) for the individual days." and have added a sentence to the table caption. All the paragraphs discussing the box modelling results have been restructured and rephrased to clarify our results.

15) Figure 5 now confuses me again about your MT scenarios. Limonene does not appear to be a fixed 20% of MTs during the day, when you are not switching off limonene. Is it 20% of emissions in your model, not concentration? You discussion made me think you had partitioned by concentration, but this looks like it contradicts that.

The limonene fraction during the day shown in Figure 5 has been double checked and it is 20% of the total coloured area (total mixing ratio).

Minor/ technical corrections and suggestions:

Line 35: format both lifetimes the same, ie., either x +/-y, or xx – yy

Done

Line 51: dependent, resulting in

Done

Line 68-69: (SOA), thereby

Done

Line 84: large variety of instruments were

Done

At lines 105-106, you might already mention that no NO3 or N2O5 results were used in this study, because they were all below DL?

Text has been added to explain that the measurements were below the LOD and therefore not used.

Line 124: suggest to mention briefly (even though you cite a reference) generally what NO measurements were corrected for.

Done

Line 178: fractional contributions (ai) from different monoterpenes as described in equation (2), where kNO3+I is the rate coefficient

Done

Line 238: Is NO3 photolysis to NO and O2 really an important atmospheric reaction?

Yes, it is up to 30% of the overall process.

Line 252: (eqn 3) error in superscript should be PNO3ANS

That has been corrected.

Line 282: missing "i" index under summation

Done

Around line 296: I think it would be clearer if you laveled the fraction alphaRO2+NO rather than just alphaRO2

Done

Line 344: daytime and nighttime from those

Done

Line 392-393: MACR), methyl vinyl ketone (…, MVK), and a-pinene, after multiple

Since both MACR and MVK are within the same () and examples of the degradation of isoprene we choose to keep the text as it was originally.

Line 397: In summertime forested environments, where

The text has been changed

Line 408-410: For PAN, MPAN … depends on the concentration of NO, … XO2 designates the sum

The text has been changed

Line 422: (R23) from 7.5 hours at 283 K to 40 minutes

Done

Line 427: an important role, depending

Done

Line 431: plots the measured mixing ratio for sumPANS (=PAN + MPAN + PPN + other PANs)

Done

Line 434: campaign result in a …magnitude, from 15 hours at 279 K to 3 minutes at 314 K as shown in Figure 7B (i.e, remove the parenthetical)

Done

Line 440-441: Figure 7), when measurements … available, have been modelled

Done

Line 450-452: and the prescribed 20% limonene monoterpene mixture described above. Note that

Done

Line 454: modelled day, 507 pptv

Done

Line 460: frequencies thus determined were

Lines 534 and 535: There are 2 instances of using the word "reactant" where I think "VOC" would be true and more clear.

Reactant has been replaced with NO and BVOC as both are required.

Figure 1: I think heat in a reaction scheme is typically indicated with a capital delta and not DELTA T. In the caption, what do you mean by OH formation in reactions of HO2 with OH? That reaction does not result in net OH formation.

ΔT has been changed to Δ in the figure and the text has been corrected to HO2 with NO giving OH.

Figure 2: Suggest to report NOx, isoprene and sum(MT) in ppbv rather than pptv. Also, could you put more of the traces on the same axes so that each individual plot is a bit taller and has a few more ticks on the y axis? For example, could combine sum MTs and isoprene on same panel, and perhaps NO2 and CH3CHO.

The unit of all the gasses have been changed to ppbv for consistency, but the panels have been kept as originally as the figure primarily serves to show what data is available.

Figure 4: suggest to switch colors to red is hotter and blue cooler (more intuitive)

The colors have been swapped in the figure.

Figure 6 panels E and F: suggest to make y axis 0 – 1500 so you can see the measurements (even though this will push the without loss case offscale sooner – you're not arguing that it's a realistic case anyway, so it's fine if it runs offscale.

The scales of Panel E and F have been adjusted as suggested.

Reviewer 2:

Summary

This paper presents ground-based observations of trace gases in a temperate forest. They use a combination of simple models to calculate production rates of total alkyl nitrates and peroxyacyl nitrates, and they examine how these depend on different oxidants and how they vary between different periods. They use the residual between modeled and observed mixing ratios to infer physical loss rates, which they imply reflect loss of these compounds via dry deposition. The writing is good and the number and style of figures is appropriate.

As detailed below, I find several major flaws in this study, specifically with regards to 1) presumption of the nature of missing sinks, 2) inadequate treatment of uncertainty, and 3) missing chemical reactions and inadequate consideration of the applicability and limitations of the model for this particular environment. Any one of these would be cause for major revisions, but in sum I regret that they justify rejection (with encouragement to resubmit).

We consider the three major flaws which the reviewer has identified in detail below. We do not explicitly equate our derived loss rate to deposition, but simply state in response to flaw-1 that deposition is a potential (partial) explanation for the short lifetimes. In this sense we have also modified the title of the manuscript. In answer to flaw-2 we have conducted a detailed analysis of uncertainties and how these relate to our derived lifetimes and impact on our conclusions. These are detailed in three new tables in the supplementary information. As to flaw-3, we do not have access to a canopy model, which the reviewer suggest may be a better choice. The use of the MCM-code should ensure a reasonably complete set of chemical reactions and rate expressions that are widely used in the box-modelling community. As the zero-D box model cannot account for transport within or beyond the clearing/canopy we have not attempted to separate the physical-losses we derived into deposition and other (i.e. transport controlled) losses. We merely propose that deposition is likely to be an important component contributing to the observed losses (especially at nighttime).

General Comments

The title is misleading. The results presented in this paper do not "indicate efficient assimilation of reactive nitrogen by the biosphere." At best, results indicate that additional sinks are sometimes needed to square a steady state model with observed concentrations. No evidence is presented that can isolate these sinks. Indeed, L554 (last paragraph of the conclusions), states that deposition is "presumably" the cause. The title should be amended to reflect defensible conclusions.

While our observations of short ON lifetimes are consistent with trapping of nitrogen by the biosphere we do agree the direct evidence of this is not available in our data. We have now modified the title to "Short lived organic nitrates in a sub-urban temperate forest: An indication of efficient assimilation of reactive nitrogen by the biosphere?". We keep some speculatory text linking the biosphere to our observations in the conclusions and mention at various positions in the text that deposition is a potential (partial) explanation for the short lifetimes, especially at nighttime.

Missing sinks: equating model-estimated "physical losses" with deposition is quite a leap.

Throughout the manuscript we have indicated that deposition *could* be responsible for the short lifetimes observed. We write for instance "The thermal losses of PANs were included in the model as described above and an additional first-order loss term (physical losses **from e.g. deposition**)". Only in the conclusions did we relate physical losses and deposition in a more concrete manner. We have now modified this to be less specific about the true nature of physical losses, see below:

"Lifetimes of organic nitrates in the forested environment are very short. A potential reason for the short lifetime is dry deposition to e.g. soil and foliar surfaces. If deposition is the cause of the short lifetimes this would be consistent with a picture of the forest ecosystem capturing essential nitrogen-containing nutrients originating from anthropogenic sources and transferring them to the biosphere."

First, a steady-state 0-D box model cannot be expected to represent all processes accurately, or even sufficiently, within a forest where there are complex interactions of emissions, radiation, and chemistry. Numerous studies using 1-D canopy models support this. The transport timescale in a typical forest canopy is a few minutes, so anything with a lifetime longer than this would be affected by those processes. Second, it is possible to estimate the expected dry deposition rates using a range of reasonable deposition velocities (Nguyen et al., 2015, and other studies). For a 1 km boundary layer and Vd = 1 – 3 cm/s, the effective deposition lifetime is 9 – 27 h, which is much longer than the 1.5 h AN lifetime estimated here. Several studies have shown that aerosol formation is important (L69). OK, that's the rant; the point is that the analysis could be improved by 1) estimating deposition loss rates based on literature, and 2) not presuming that "physical loss" means deposition.

We have discussed potential other loss processes such as transport via e.g. canopy venting and shown that the physical losses of ANs at day and nighttime needed to explain the observations are the same within the uncertainties. As nighttime transport out of the canopy is expected to be greatly reduced this is unlikely to be the major reason for short calculated lifetimes.

We did not calculate the deposition velocities because we also considered that the link between our lifetime calculations and deposition is indirect, but only suggested that dry deposition is a possible explanation. This concurs with the reviewer's comment "…. equating model-estimated "physical losses" with deposition is quite a leap." We note that effective deposition lifetimes of 9-27 h mentioned by the reviewer are not consistent with the rapid depletion of ozone, which sometimes led to lifetimes of a few hrs (Andersen et al., 2024), which could not be attributed to chemical processes and is also consistent previous observations of short ON lifetimes.

Aerosol formation cannot account for the short lifetimes of organic nitrates as no increase in particulate nitrate was observed. This is already mentioned in the text.

Nighttime OH: L261 states that nighttime OH can be explained by ozonolysis, but analysis is never shown to support this. Later, L346 states that a major difference between this study and a previous one is the nighttime OH data. This is not the first time nocturnal OH has been observed, and I suspect the jury is still out on the extent to which this might be a measurement artifact. Given this appears to be important, the

paper should include some analysis of whether or not the observed nighttime OH is consistent with known chemistry.

We now write "The most important nighttime source of OH radicals is generally believed to be the reaction between unsaturated VOCs and ozone e.g. (R11-R12). As will be shown in a forthcoming publication, the nighttime OH levels are broadly consistent with measured OH-reactivity, HO2-recycling via reaction of HO2 with NO and production via the ozonolysis of terpenoids". We have no reason to believe that the nighttime OH measurements are biased as rigorous corrections of interfering background signals was performed during the campaign.

Uncertainty: Measurement uncertainties are not propagated. This is especially important for any calculations using OH. Figures 6 C-D could use error bars or something in the text discussing how uncertain that balance is. Figure 8 A-B would also benefit from an uncertainty estimate on the modeled PAN – at least from known errors. This doesn't have to be fully rigorous to be useful.

A detailed uncertainty analysis for the ANs production, fractional contribution and lifetime has been added to the paper.

Specific Comments

Sect. 3.2.1: What is the accuracy of the reactive N measurements?

LOD and total uncertainties of the NO2, PANs and ANs measurements have been added to the text.

L124: Corrected for what?

Text has been added.

Sect. 3.2.2: how confident are you that the observed OH is representative of the average for the forest canopy or the larger environment that would contribute to VOC oxidation? If you are in a forest, presumably there is significant shading.

Our analysis is based on observations in a clearing, which is almost always the case for trace-gas measurements in forests, and which might not be completely transferable to non-cleared woodland. We now write "Further as discussed by Dewald et al. (2024) the relative contributions of e.g. OH and NO3 would be significantly modified to favour NO3 if we consider the greatly reduced photolysis frequencies of NO3 and OH precursors in non-cleared parts of the forest.". In the absence of in-forest observational data this is the best we can do.

Sections 4.1 – 4.3 present theory that is used to analyze the results, not the results themselves. As such, this should be moved to Methods or a "Theory" section.

We feel that having the theory directly before discussing the results avoids repetition and would therefore like to keep it where it is.

Sect. 4.1 does not include N2O5. This seems like a major omission, especially considering Sect. 3.2.1 states N2O5 was measured. This needs to be included in the NO3 steady state calculation (including heterogeneous loss of N2O5), or a quantitative argument needs to be made on why it can be excluded.

The N2O5 measurements were below the LOD of the instrument almost every night and on the nights where small increases in N2O5 were observed it only lasted for a short time period. Text has been added to explain this in the instrumental part of the paper.

L280: What is the basis for the generic rate coefficients?

We omitted to describe this but now write:
"$\beta$ was calculated using the measured $XO_2$ ($HO_2$+$RO_2$) and NO together with a generic rate coefficient for reaction (R8), $8 \times 10^{-12}$ $cm^3$ molecule$^{-1}$ s$^{-1}$, and a generic rate coefficient for the combination of reactions (R9) and (R10), $k_{9/10} = 1 \times 10^{-11}$ $cm^3$ molecule$^{-1}$ s$^{-1}$ (IUPAC, 2024; Lightfoot et al., 1992). $k_8$ was set to this value because a vast majority of organic peroxy radicals react with NO with a rate coefficient of $8 \pm 1 \times 10^{-12}$ $cm^3$ molecule$^{-1}$ s$^{-1}$ (IUPAC, 2024; Lightfoot et al., 1992). Since we do not have separate measurements of $RO_2$ and $HO_2$ an effective rate coefficient of $1 \times 10^{-11}$ $cm^3$ molecule$^{-1}$ s$^{-1}$ was chosen. This was derived by considering the MCM rate coefficients ($2.3 \times 10^{-11}$ $cm^3$ molecule$^{-1}$ s$^{-1}$ at 298 K) for reaction (R10) and 2-3 $\times$ $10^{-12}$ $cm^3$ molecule$^{-1}$ s$^{-1}$ for reaction between isoprene derived peroxy radicals and other $RO_2$ and considering the geometric mean."

L376: This is also consistent with another very different analysis of isoprene nitrates, which found lifetime of < 2h with losses dominated by aerosol uptake (Wolfe et al, 2015, Table S5).

True. The reference has been added with a brief description.

L378: a potential implication of XO2 over-prediction is that OH is too high. How would reduced OH alter your estimated AN lifetime? At a minimum, this should be considered from the standpoint of the presumed 25% uncertainty in observed OH.

A detailed uncertainty analysis of uncertainty in deriving the ANs production rates and lifetime (including the uncertainty in OH and XO2) have now been added to the SI and the major results are described in the text.

L405: MPA radical can also isomerize (Crounse et al., 2012). Presumably this has a relatively minor impact on modeled PANs.

We have now mentioned the isomerization in the text that described the degradation of the methacrolein radical. We also indicate that MPAN is a small contribution to the total PANs later in the SI.

L465: This is also consistent with analysis of PAN fluxes above another forest (Wolfe et al., 2009), although in that study they found a downward flux driven by warmer temperatures at the surface (meaning that canopy transport was a net source of PAN).

We now write "The 40% contribution of thermal loss appears to be inconsistent with the observations of Wolfe et al. (2009) who showed through flux measurements that thermal losses of PAN were dominant.".

Technical Comments

L29: Define "ACROSS"

Done

L53: capitalize IUPAC

Done

L146: suggest striking this sentence. I believe it is no longer acceptable to refer to an unwritten paper, especially since they often remain unwritten.

We have extended some of the text describing the HOx measurements to provide the reader with more information, but a detailed analysis of the OH sources and sinks and HOx modelling is beyond the scope of the present manuscript.

L160: define LISA

Done

L230" delete "i.e. . . . terms."

Done

L269: OH addition, which is the dominant channel for alkenes, does not lead to H2O formation.

H2O has been removed.

L378: change "above" to "Sect. 3.3."

Done

L498: Fig. 8

Done

References from reviewer 2:

Crounse, J. D., et al.: On the atmospheric fate of methacrolein: 1. Peroxy radical isomerization following addition of OH and O2, Journal Of Physical Chemistry A, 116, 5756–5762, https://doi.org/10.1021/jp211560u, 2012.

Nguyen, T. B., Crounse, J. D., Teng, A. P., St. Clair, J. M., Paulot, F., Wolfe, G. M., and Wennberg, P. O.: Rapid deposition of oxidized biogenic compounds to a temperate forest, Proceedings of the National Academy of Sciences, 112, E392–E401, https://doi.org/10.1073/pnas.1418702112, 2015.

Wolfe, G. M., et al.: Eddy covariance fluxes of acyl peroxy nitrates (PAN, PPN and MPAN) above a Ponderosa pine forest, Atmospheric Chemistry And Physics, 9, 615–634, 2009.

Wolfe, G. M., et al.: Quantifying sources and sinks of reactive gases in the lower atmosphere using airborne flux observations, Geophysical Research Letters, 42, 8231–8240, https://doi.org/10.1002/2015GL065839, 2015

References used in replies:

Andersen, S. T., McGillen, M. R., Xue, C., Seubert, T., Dewald, P., Türk, G. N. T. E., Schuladen, J., Denjean, C., Etienne, J. C., Garrouste, O., Jamar, M., Harb, S., Cirtog, M., Michoud, V., Cazaunau, M., Bergé, A., Cantrell, C., Dusanter, S., Picquet-Varrault, B., Kukui, A., Mellouki, A., Carpenter, L. J., Lelieveld, J., and Crowley, J. N.: Measurement report: Sources, sinks, and lifetime of NOx in a suburban temperate forest at night, Atmos. Chem. Phys., 24, 11603-11618, 10.5194/acp-24-11603-2024, 2024.

Dewald, P., Seubert, T., Andersen, S. T., Türk, G. N. T. E., Schuladen, J., McGillen, M. R., Denjean, C., Etienne, J. C., Garrouste, O., Jamar, M., Harb, S., Cirtog, M., Michoud, V., Cazaunau, M., Bergé, A., Cantrell, C., Dusanter, S., Picquet-Varrault, B., Kukui, A., Xue, C., Mellouki, A., Lelieveld, J., and Crowley, J. N.: NO3 reactivity during a summer period in a temperate forest below and above the canopy, Atmos. Chem. Phys., 24, 8983-8997, 10.5194/acp-24-8983-2024, 2024.

IUPAC Task Group on Atmospheric Chemical Kinetic Data Evaluation, (Ammann, M., Cox, R.A., Crowley, J.N., Herrmann, H., Jenkin, M.E., McNeill, V.F., Mellouki, A., Rossi, M. J., Troe, J. and Wallington, T. J.). Last access April. 2024: https://iupac.aeris-data.fr/, last

Lightfoot, P. D., Cox, R. A., Crowley, J. N., Destriau, M., Hayman, G. D., Jenkin, M. E., Moortgat, G. K., and Zabel, F.: Organic peroxy radicals - kinetics, spectroscopy and tropospheric chemistry, Atmospheric Environment, Part A: General Topics, 26, 1805-1961, 1992.

Wolfe, G. M., Thornton, J. A., Yatavelli, R. L. N., McKay, M., Goldstein, A. H., LaFranchi, B., Min, K. E., and Cohen, R. C.: Eddy covariance fluxes of acyl peroxy nitrates (PAN, PPN and MPAN) above a Ponderosa pine forest, Atmospheric Chemistry and Physics, 9, 615-634, 2009.

---

## Author Response (AR2)

We thank the reviewer for the additional comments. Below we have written the comments in black and our replies in red.

Thanks to the authors for addressing many reviewer comments and suggestions. Two final suggestions to strengthen the paper:

(1) The new figure showing the measurement site is very helpful, but since many questions are about inlet heights, it would be great if you could also add a "side view" with inlet heights on the mast / containers for the relevant instruments shown, as well as the height of the trees relative to those inlets. This can't easily be seen via the top down image.

A side view has been added to the supplementary information.

(2) Please use your box model to predict the formation of OH from O3 + BVOC to check whether it is consistent with the nighttime measurements and comment on this clearly, alongside the 25% uncertainty. This remains surprising and it seems it would be possible in your model to check this. Thank you!

While we understand the reviewers concern about high OH levels at nighttime, a rerun of our model with unconstrained OH is unlikely to either prove or disprove the quality of the OH measurements. Problems which the MCM mechanism has in predicting OH levels in the presence of biogenic VOCs is documented (see eg. a recent publication in ACP: https://doi.org/10.5194/acp-22-8497-2022). The research group who took the OH measurements are presently examining sensitivities to various parameters such as eg. NO which controls efficiency of OH recycling and its levels both day and night. This requires substantial modification of our MCM code which is beyond the scope of the present manuscript. We thus prefer to trust the measurements and expect that a future publication dealing with model-measurement comparison of HOx will shed light on the high OH levels observed.